# Subtype-Selective Peptide and Protein Neurotoxic Inhibitors of Nicotinic Acetylcholine Receptors Enhance Proliferation of Patient-Derived Glioblastoma Cell Lines

**DOI:** 10.3390/toxins16020080

**Published:** 2024-02-02

**Authors:** Elena Gondarenko, Diana Mazur, Marina Masliakova, Yana Ryabukha, Igor Kasheverov, Yuri Utkin, Victor Tsetlin, Mikhail Shahparonov, Denis Kudryavtsev, Nadine Antipova

**Affiliations:** 1Department of Molecular Neuroimmune Signaling, Shemyakin-Ovchinnikov Institute of Bioorganic Chemistry of Russian Academy of Sciences, 117997 Moscow, Russia; gondarenkoelena@ibch.ru (E.G.); iekash@ibch.ru (I.K.); vits@ibch.ru (V.T.); kudryavtsev@ibch.ru (D.K.); 2Department of Functioning of Living Systems, Shemyakin-Ovchinnikov Institute of Bioorganic Chemistry of Russian Academy of Sciences, 117997 Moscow, Russia; mazur@ibch.ru (D.M.); masliakova@ibch.ru (M.M.); ryabukha_yana@ibch.ru (Y.R.); shakhparonov@ibch.ru (M.S.); antipova@ibch.ru (N.A.); 3Department of Biology and General Genetics, I.M. Sechenov First Moscow State Medical University, 119048 Moscow, Russia

**Keywords:** glioblastoma multiforme, patient-derived cell lines, nicotinic acetylcholine receptors, snake neurotoxins, α-conotoxins, proliferation

## Abstract

Glioblastoma multiforme (GBM) is the most aggressive type of brain cancer, with a poor prognosis. GBM cells, which develop in the environment of neural tissue, often exploit neurotransmitters and their receptors to promote their own growth and invasion. Nicotinic acetylcholine receptors (nAChRs), which play a crucial role in central nervous system signal transmission, are widely represented in the brain, and GBM cells express several subtypes of nAChRs that are suggested to transmit signals from neurons, promoting tumor invasion and growth. Analysis of published GBM transcriptomes revealed spatial heterogeneity in nAChR subtype expression, and functional nAChRs of α1*, α7, and α9 subtypes are demonstrated in our work on several patient-derived GBM microsphere cultures and on the U87MG GBM cell line using subtype-selective neurotoxins and fluorescent calcium mobilization assay. The U87MG cell line shows reactions to nicotinic agonists similar to those of GBM patient-derived culture. Selective α1*, α7, and α9 nAChR neurotoxins stimulated cell growth in the presence of nicotinic agonists. Several cultivating conditions with varying growth factor content have been proposed and tested. The use of selective neurotoxins confirmed that cell cultures obtained from patients are representative GBM models, but the use of media containing fetal bovine serum can lead to alterations in nAChR expression and functioning.

## 1. Introduction

Glioblastoma multiforme (GBM) is the most aggressive and lethal type of tumor of the central nervous system (CNS), with a median patient survival of 15 months from the time of diagnosis [1]. Unfortunately, current standard of care, including surgical resection, radiotherapy, and oral administration of the DNA-alkylating drug temozolomide, show limited efficacy in GBM treatment [2].

The Cancer Genome Atlas (TCGA) proposed four main GBM subtypes based on their genomic features: proneural, mesenchymal, neural, and classical [3,4]. Several studies identified the parameters that can be used to characterize various subtypes of the tumor [4,5,6]. Recent data show that a proneural-to-mesenchymal transition (PMT) may occur at different stages of the disease. Tumor transformation into mesenchymal subtype GBM stem cells is associated with more aggressive phenotypes, treatment resistance, and tumor recurrence [7,8].

The microenvironment of GBM cells also plays an important role in the development, recurrence, and resistance of brain cancer [9]. GBM cells undergo dynamic and reversible transitions in response to changes in the microenvironment, which in turn changes the spatial and temporal phenotypic heterogeneity of the tumor [10]. Numerous studies show that interaction with neurons, as well as direct and indirect consequences of neuronal activity, can influence tumor progression [11,12].

It has been shown that neurons affect tumor growth and progression by forming functional synapses with glioma cells [13,14]. For instance, GABA-mediated synaptic transmission inhibits GBM growth, whereas active release of glutamate increases cell proliferation [15]. Other signaling molecules transmitted between glioma cells and normal neurons are brain neurotrophic factor, neuroligin-3, and dopamine [16]. The cholinergic projections are also known to innervate regions where GBM develops, releasing the neurotransmitter acetylcholine (ACh) [17].

Several subtypes (mainly containing α1, α7, α9, and α10 subunits) of nicotinic acetylcholine receptors (nAChRs) effectively permeate calcium ions [18]. It has been confirmed that GBM cells express certain types of nAChRs, and the functionality of these receptors has been demonstrated [17,19]. The expression of the majority of nAChR genes in GBM cells is at a lower level compared to control samples. In patient-derived cell lines, the most frequently expressed nAChR subunit mRNAs are *CHRNA5* (which encodes α5 subunit) and *CHRNA9* (α9), and at a lower level, *CHRNA7* (α7) and *CHRNA10* (α10). In the model U87MG line, the expression of muscle nAChR subunits *CHRNA1* (α1) and *CHRNB1* (β1) was also shown [17,19].

Intracellular calcium has been implicated in various signaling pathways that affect tumor progression. Several studies suggest that GBM proliferation, migration, and invasion change in the presence of nAChR ligands [17,19]. These effects are predominantly mediated by α7 and α9 nAChRs [17,19]. Some GBM model lines such as U251 proliferate in the presence of the nAChR agonist acetylcholine [17]. Complementary to that, 10^−7^ M nicotine increases the proliferation of the U87MG and GBM5 lines [19]. The effect of nicotine is blocked by both α-bungarotoxin (α-Bgt) and methyllycaconitine (MLA), which unselectively inhibit muscle, α7, and α9 nAChRs, as well as by the α7-selective antagonist α-conotoxin analogue ArIB (V11L; V16D) and the α9-selective antagonist α-conotoxin RgIA [19].

Furthermore, GBM can easily develop radio- and chemoresistance due to the extreme heterogeneity of these tumors [20]. Different GBM cell subpopulations retain distinct genetic profiles, biomolecular markers, and functional properties [21]. In this regard, it is crucial to choose the right models to study molecular and biochemical features and to test the effects of potential therapeutic substances. Patient-derived cancer stem cell (CSC)-enriched cultures grown in serum-free media are the most accepted standard for studying the biology of GBM in vitro [21]. However, the same patient-derived cultures grown under standard in vitro conditions with serum-supplemented media have gene expression profiles that are dramatically different from either CSCs or the original tumors [22].

In this paper, we explore nAChR subtypes in patient-derived and in model U87MG GBM cell lines using a range of methods, including real-time PCR, fluorescent calcium imaging, electrophysiology, and AlamarBlue proliferation assay. In our work, we studied the effect of cultivation conditions of patient-derived cultures on the expression and functional activity of different nAChR subtypes, as well as the effect of nAChR ligands on cell proliferation under different growing conditions. To distinguish the effects of different nAChRs, a variety of ligands was used in the current study (Figure 1). Among them, epibatidine activates all nAChR subtypes except α9, for which it is an inhibitor. In order to determine which specific nAChR subtype activation is associated with changes in GBM cell proliferation, the α1, α7, and α9 subunit-selective antagonists azemiopsin, α-conotoxin [A10L]PnIA, and α-conotoxin RgIA, respectively, were added to cells, along with epibatidine, acetylcholine, or choline or without any externally added agonists. To our knowledge, this is the first report on the use of several nAChR subtype-selective peptide neurotoxins in studying primary GBM.

## 2. Results

### 2.1. Transcriptomics Database Analysis

The Ivy Glioblastoma Atlas Project (Ivy GAP) database was used for the analysis [23]. Briefly, the database contains anatomic-aware data on gene expression in 42 glioblastoma tumors from a 41-patient cohort. A matrix with RNA sequencing data of the nAChR genes *CHRNA1*–7, *CHRNA9*–10, *CHRNB1*–4, *CHRND*, *CHRNE*, and *CHRNG* and the muscarinic acetylcholine receptor (mAChR) gene *CHRM1*–5 was obtained from the database (Figure 2).

Gene expression data were divided into seven different zones. The zones reflect the relationship between the anatomical features of GBM and the gene expression profile. As can be seen in Figure 2, significant differences in the expression level of acetylcholine receptor genes were observed between samples for several genes. In particular, the expression of the *CHRM1*, *CHRNA2*, *CHRNA4*, *CHRNA6*, *CHRNA7*, and *CHRNB2* genes was increased in samples from the tissue of the anterior edge of the tumor. The expression of the *CHRNA9* gene was reduced in the same region but increased in the perinecrotic zone and pseudopalisading cells (Figure 2). It is worth noting that progressing glioblastoma lesions are characterized by markedly elevated choline levels [24,25]. Thus, the expression profiles of choline-sensitive muscle (α1β1δε), α7, and α9α10 nAChRs are of a particular interest [26,27,28].

### 2.2. Real-Time qPCR Detection of nAChR Subunits

The results of the RNA-Seq database analysis from Ivy GAP demonstrate differential expression of choline-sensitive nAChRs (α7 and α9) in different morphological zones of the tumor. Apart from brain lesions, choline is present in various body fluids, e.g., the blood plasma concentration of choline in a recent study was estimated using LC-MS/MS to be as high as 11.4 ng/μL, which corresponds to ~100 μM [29], although NMR data showed concentrations of up to 28.9 μM of choline in the blood serum of healthy individuals [30]. In view of these literature data, we decided to study the expression of nAChR genes via patient-derived GBM microsphere cultures [31] using real-time PCR after culturing cells in two types of medium: without fetal bovine serum (but with bFGF and EGF growth factors and NS-21 neuronal vitality supplementation) and with FBS (Figure 3).

Most of the lines studied showed higher levels of nAChR gene expression after cultivation under serum-free conditions compared to cultivation under serum-supplemented conditions (Figure 4A).

Interestingly, α5 and α9 nAChR mRNA were detected in all of the lines studied under both serum-containing and serum-free conditions (Figure 4B,C). The expression of the α7 nAChR gene was observed in all cell lines cultivated in both FBS-supplemented and serum-free medium, except for the 019 proneural line in serum-free conditions. The U87MG cell line showed similar expression profiles in both media variants; however, the expression of the α7 nAChR subunit gene increased in serum-free medium. The 011 proneural line showed expression of almost all nAChR subunits at the mRNA level (except for α4 and β2 in serum-free medium and α4 in serum-containing medium), whereas the 019 proneural line expressed α1, α5, and α9 nAChR genes in serum-free conditions and α7 nAChR gene expression was activated in medium with FBS in addition to other nAChR subunits. Interestingly, the response of proneural lines to cultivation in medium with FBS was manifested by the emergence of the expression of some subunits that were not observed under serum-free conditions. Cell lines of the mesenchymal subtype showed the presence of α5, α7, and α9 nAChR in both media variants. Mesenchymal subtype line 022 demonstrated additional expression of α1, α3, α10, and β2 nAChR subunits genes (except for α10 in serum-containing medium).

### 2.3. Surface nAChR Expression Confirmed by Fluorescently Labeled α-Bgt

To visualize functional nAChRs on the cell membrane, labeling with the fluorescent Alexa Fluor 555-α-bungarotoxin (50 nM, α-Bgt-Alexa 555) was carried out. α-Neurotoxin α-Bgt from multi-banded krait *Bungarus multicinctus* venom is a competitive antagonist of α7, α9 and muscle nAChRs and binds to these receptors with high affinity. The intensity of α-Bgt-Alexa 555 staining of mesenchymal GBM cells significantly exceeded the background level (Figure 5A). To confirm the specificity of the nAChR labeling by α-Bgt-Alexa 555, a 20-fold molar excess of d-tubocurarine (d-TC), another competitive antagonist of nAChRs, was added and a significant reduction in receptor labeling was observed (Figure 5B).

Thus, labeling with fluorescent α-Bgt confirmed nAChR gene expression and translation in the mesenchymal cell line. Since the expression of the α1 nAChR subunit was not detected on the 067 cell line via RT-PCR, the fluorescent ligand binding can be attributed to the nAChR of either α7 or α9.

### 2.4. Study of nAChR Functional Activity by Calcium Imaging and Electrophysiology

Fluorescent single-cell calcium imaging was performed to assess whether the nAChR gene expression observed via RNA-Seq database analysis from Ivy GAP and RT-PCR (Figure 2) correlated with the yield of functional nAChRs. Since mRNA expression of α1, α7, and α9 subunits forming calcium-conducting nAChRs was detected (Figure 4), functional tests were performed with ligands selective for these subunits.

Using the calcium-imaging technique with an epifluorescent microscope, it was found that all cell lines were responsive to the application of 10 μM ACh. To demonstrate which particular receptor subtype was activated, nAChR inhibitors were used in combination with ACh. GBM cells were preincubated either with azemiopsin, a selective competitive antagonist of muscle-type nAChR; with α-conotoxin [A10L]PnIA, which selectively blocks α7 nAChR; or with α-conotoxin RgIA, which inhibits α9 nAChR. Thus, it was observed that, in U87MG cells grown on FBS-supplemented medium, α-conotoxin [A10L]PnIA selective for α7 nAChR suppressed an ACh-stimulated increase in intracellular Ca^2+^ levels while azemiopsin and α-conotoxin RgIA were inactive (Figure 6 and Figure 7A). This means that functionally active α7 nAChR was present on the U87MG cells cultivated on FBS-supplemented medium (Figure 6 and Figure 7A).

Similar experiments with ACh-evoked Ca^2+^ entry inhibition were carried out with patient-derived GBM cell lines. One of the proneural GBM cell lines, namely, 011, did not show signs of acetylcholine-evoked calcium increase that could have been inhibited by muscle, α7, or α9 nAChR inhibitors, suggesting a limited presence of functioning receptors of these subtypes in cells cultivated in serum-free medium (Figure 7B, right panel). It should be noted that for the cells grown on the FBS-supplemented medium, the Kruskal–Wallis test detected significant (*p* < 0.01) differences in calcium increase amplitudes between control and inhibitor-treated cells (Figure 7B, left diagram). The pairwise Dunn’s test, however, did not detect significant differences between groups, suggesting that the effect of selective inhibitors, if present, is of limited value. It was found that in another proneural cell line (019) grown on a serum-supplemented medium, the response to ACh was inhibited by α-conotoxins [A10L]PnIA and RgIA as well as by azemiopsin (Figure 7C, left panel). This means that α7, α9, and/or muscle-type functional nAChRs are present on the surface of the cells. Therefore, the genes that encode subunits of these receptors and that are detected earlier by qPCR in this cell line (Figure 4C) are translated into active receptors. This cell line, when grown on serum-supplemented medium, adhered well to the coverslip used in the patch-clamp technique, and whole-cell dose-dependent currents were recorded in response to α7 nAChR-selective agonist PNU-282987 in the presence of positive allosteric modulator PNU-120596, confirming the functional cell-surface expression of α7 nAChR (Figure 7D). However, no functionally active receptors were observed on the same line when cultivated on a serum-free medium with growth factors. On the contrary, both patient-derived cultures of the mesenchymal subtype, which were grown on a serum-free medium, expressed α7 nAChR, although no nAChR was detected when these lines were cultivated on serum-supplemented medium (Figure 7E,F).

To explore the selectivity of the effects observed in the calcium-imaging experiments, the wider panel of nAChR ligands was deployed on the 022 line (Appendix A). Interestingly, nicotine, PNU-282987, and methyl ester of D-6-Iodohypaphorine (6ID) and D-6-nitrohypahporine amide (6NAM) evoked calcium responses. D-tubocurarune effectively inhibited all responses to solo agonists but was not effective against PNU-282987 or 6NAM when co-applied with positive modulator PNU-120596 (pnu12). Three-finger toxin α-cobratoxin (α-CTX), being a selective muscle, α7, and α9 nAChR inhibitor, diminished responses to nicotine and 6NAM when PNU-120596 was not co-applied. These results speak in favor of α7 nAChR presence in the 022 line and are in good agreement with the data presented in Figure 7E.

### 2.5. GBM Cell Proliferation after the Exposure to nAChR Agonists and Antagonists

As shown in the previous sections, the GBM cell lines produced functionally active calcium-conducting nAChRs containing α1, α7, and α9 subunits, and expression of the particular receptor subtype depended on the cell line and cultivation condition. We decided to study how subtype-selective nAChR ligands and cultivation conditions affect the proliferation of different GBM cell lines. To evaluate these effects, the tests were performed under two condition variants: serum-free medium and FBS-containing medium. The influence of nAChR ligands on GBM cell proliferation was studied using the resazurin-based colorimetric method (AlamarBlue). The effects of varying concentrations of epibatidine in combination with muscle and α7 nAChR-specific antagonists were studied. In the AlamarBlue proliferation assay, the non-selective nAChR agonist epibatidine was used to eliminate the influence on mAChR.

There was practically no effect of epibatidine at different concentrations on the studied cell lines. Only at 10 nM did epibatidine significantly increase the proliferation of proneural patient-derived GBM cell line 011 grown on serum-free medium (Figure 8B). The addition of α7 nAChR antagonist [A10L]PnIA along with epibatidine led to an increase in cell proliferation in both FBS-supplemented (Figure 8A,E,G) and serum-free medium (Figure 8F,H,J) in four of five analyzed lines, namely, 011, 022, 067, and U87MG.

The proliferation of the model GBM line U87MG grown on FBS-supplemented medium increased in response to the combination of [A10L]PnIA and 10 nM epibatidine. Also, the proliferation of U87MG grown on serum-free medium increased in response to azemiopsin in combination with epibatidine at 10 and 30 nM. Epibatidine is an nAChR agonist, whereas azemiopsin is a selective muscle-type receptor inhibitor. In proliferation tests, the effects of both agonist-mediated activation of receptors and their inhibition by selective antagonists were investigated to clarify whether the antagonist counteracts possible effects of the agonist.

It should be noted that proneural GBM cell lines 011 and 019 responded differently to nAChR antagonists. The proliferation of cell line 011 increased under certain conditions (especially on serum-containing medium), whereas in line 019, the cell proliferation level did not change in any of the experiments (Figure 8A–D). For instance, cell line 011 grown on FBS-supplemented medium demonstrated an increase in cell proliferation under the influence of [A10L]PnIA in the presence of epibatidine (Figure 8A).

Cells of the mesenchymal subtype (022 and 067), which are characterized by a high level of stemness and a more aggressive phenotype, also showed different results. In particular, cell line 067 did not exhibit increased proliferation in response to exposure to the muscle-type nAChR antagonist azemiopsin (the small and concentration-independent effect in the presence of 0.03 μM epibatidine only on FBS-supplemented medium should probably be considered an outlier; see Figure 8G), whereas proliferation of cell line 022 increased in both FBS-supplemented and serum-free medium (Figure 8E,F).

The observed difference can be explained by the extreme heterogeneity of GBM cells, meaning that cells belonging to the same phenotype may have different properties and respond differently to the action of different agents. Thus, almost no direct influence of epibatidine on the proliferation of GBM cells was detected and the addition of nAChR inhibitors showed pro-proliferative effects.

Since epibatidine is an agonist of a broad spectrum of nAChR subtypes, selective α7 nAChR agonist PNU-282987 and α-CTX were also tested on the 022 line to check the effect selectivity. In accordance with the epibatidine/toxin results presented in Figure 8, no significant influence of PNU-282987 on the proliferation of cell line 022 was observed at concentrations from 20 nM to 6 μM. The addition of 1 μM α-CTX to cells incubated with PNU-282987 increased cell proliferation. The effect was dose dependent and was observed only at low agonist concentrations (Appendix A).

Since the addition of antagonists together with an agonist induced proliferation of GBM cells, we decided to test the effect the antagonists alone would produce and repeated the AlamarBlue assay using α1, α7, and α9 nAChR antagonists but without the agonist. Again, the incubation was performed in two media: with and without FBS (Figure 9).

In the U87MG cell line as well as in the proneural lines 011 and 019, the proliferation increased in response to exposure to nAChR antagonists in FBS-supplemented medium, whereas under serum-free conditions, cell proliferation remained at the control level (Figure 9A–C). The U87MG cell line showed increased cell proliferation in response to all nAChR antagonists used; the greatest increase in proliferation was observed in the presence of α-conotoxin RgIA, an α9-selective antagonist (Figure 9A). Cell line 011 responded significantly only to the α9 nAChR antagonist, and in cell line 019, proliferation increased after exposure to azemiopsin and α-conotoxin [A10L]PnIA, which are selective antagonists of α1- and α7-containing nAChRs, respectively (Figure 9B,C). Cells of the mesenchymal lines 022 and 067 showed no significant changes in proliferation in either media variant (Figure 9D,E).

To gain an understanding of choline’s role in the nAChR antagonist-mediated increase in proliferation, we performed two more AlamarBlue tests. In the first one, we used serum-free medium with the addition of growth factors (bFGF and EGF) and a neuronal vitality supplement (NS-21). The effects of varying concentrations of choline in combination with antagonists were studied as in the previous experiments (Figure 10). The model line U87MG showed no significant changes in cell proliferation (Figure 10A), whereas the proliferation of line 011 increased in the presence of RgIA and [A10L]PnIA in comparison to choline (Figure 10B) and line 019 showed increased proliferation in response to all ligands at a choline concentration of 1 mM as well as in response to azemiopsin, an α1 nAChR antagonist, at a choline concentration of 10 mM (Figure 10C). Line 022 showed no significant changes in cell proliferation (Figure 10D), whereas line 067 showed an increase in cell proliferation at choline concentrations of 0.1 to 10 μM in response to α7 and α9 nAChR antagonists (Figure 10E). It is worth noting that of all the lines studied, line 067 was the most aggressive and resistant to therapy. These results suggest that the action of choline present in FBS may influence cell growth through persistent activation of muscle, α7, and α9 nAChRs.

As already mentioned, line 067 was the most aggressive among the lines tested, and its proliferation increased in response to the action of muscle, α7, and α9 nAChRs antagonists in a wide range of choline concentrations. The influence of growth factors on the effects of nAChRs on proliferation was further investigated on the 067 line under different cultivation conditions (Figure 11). Tests using epibatidine alone or in combination with nAChR inhibitors were performed. The effects of antagonists without an agonist were also studied. Line 067 showed no significant changes in cell proliferation in the NS-21-supplemented medium without the addition of bFGF or EGF (Figure 11A). There was a slight increase in cell proliferation in the medium without EGF but with the addition of bFGF and NS-21 in response to the α7 nAChR antagonist in the presence of epibatidine (Figure 11B). Interestingly, epibatidine strongly decreased cell proliferation in the medium containing no growth factors (Figure 11C). Muscle nAChR antagonist azemiopsin diminished the effect of low concentrations of epibatidine (0.03 and 0.1 µM, Figure 11C). Interestingly, the α7 nAChR antagonist [A10L]PnIA rescued GBM cells from growth suppression by 0.03, 0.1, 0.3, and 1 µM epibatidine (Figure 11C). These results may indicate that, in the absence of all growth factors, the activation of α7 and muscle-type nAChR significantly reduced cell proliferation. A slight increase in proliferation in response to the α1 nAChR antagonist azemiopsin was observed in NS-21-supplemented medium without EGF or bFGF (Figure 11D). Cell proliferation was significantly increased in response to the α7 and α9 nAChR antagonists with no external agonists applied in medium without EGF but with the addition of bFGF and NS-21 (Figure 11E). However, there was no change in cell proliferation in medium without any growth factors (Figure 11F).

## 3. Discussion

In recent years, a wide range of methods has been applied to study GBM, which made it possible to understand certain mechanisms of tumor development [32]. Cancer cell lines have been used as the standard both for studying the biology of human tumors and as preclinical screening models of potential therapeutic agents [32]. However, it is becoming increasingly clear that phenotypic characteristics and a variety of genetic aberrations found in cancer cell lines, which repeatedly occurs in vitro, for instance, often bear little resemblance to those found in the corresponding human patient-derived tumor. This may have led to some significant misinterpretations regarding the significance of aberrant signaling pathways within cell lines compared to patient-derived tumors. GBM tumor cell populations have a variety of functional and molecular biological features, which makes this type of tumor extremely heterogeneous [20]. In particular, glioblastoma stem cells (GSCs), which are characterized by increased resistance to chemotherapy and radiotherapy, are present in cell populations [33].

The immortalized U87MG cell line has been used in many laboratories worldwide, but due to genetic drift in FBS-supplemented medium and successive passages, the cells with the highest proliferative potential are selected, reducing the genetic heterogeneity inherent to the original tumors, which can affect the reproducibility of studies [34]. The cell cultures obtained from patients are more representative models, but the use of media containing fetal bovine serum (FBS) can lead to loss of the GSC subpopulation [34], making the tumor homogeneous. Among other things, the presence of GSCs is closely linked to the formation of spheroids, called neurospheres, in tumor cell cultures of neural-originating tumors [35]. It was previously shown that glioma cells cultured in either serum-containing or serum-free medium showed profound biological differences, particularly, in terms of the increased differentiation of CSCs and the loss of tumor heterogeneity [22]. One approach to the cultivation of cells as microspheres to prevent the loss of tumor heterogeneity is to use serum-free medium containing basic fibroblast growth factor (bFGF), epidermal growth factor (EGF), and a neuronal viability supplement (B27 or NS-21) [36].

In the current work, the effects of the cultivation conditions on the expression of several nAChR subtypes in different patient-derived GBM cell lines were investigated. Using nAChR subtype-selective ligands, we also studied the roles played by these receptors in the proliferation of GBM cell lines. The presence of nAChR gene expression of several nAChR subunits in patient-derived cultures of samples 011, 019, 067, and 022 at the mRNA level was demonstrated. Under different cell culture conditions (FBS-supplemented vs. NS-21/EGF/bFGF-containing), patient-derived GBM cultures from different human sources showed distinct profiles of nAChR gene expression (Figure 4). Of note, α5 and α9 nAChR mRNA were detected in all GBM lines tested. GBM line 011 expressed the β2 nAChR subunit gene exclusively when cultured on FBS-supplemented medium, and mRNA of the α9 nAChR subunit was significantly increased in FBS-supplemented medium in contrast to serum-free medium. Lines 011, 019, and 022 showed a significant increase in muscle nAChR α1 subunit gene expression in serum-free medium in contrast to FBS-supplemented medium. Gene-encoding α7 nAChR was reduced in serum-free medium compared to FBS-supplemented medium in the U87MG model cell line, but in the 019 line, the expression of the same nAChR gene was observed only on FBS-supplemented medium, whereas other lines expressed this receptor independently of culturing conditions. Overall, mesenchymal lines 022 and 067 showed less dependence on nAChR gene expression on the composition of the medium (FBS-supplemented or serum-free medium) than proneural lines 011 and 019. The presence of nAChR in mesenchymal line 067 was confirmed by specific binding of fluorescent ligand Alexa Fluor 555 α-bungarotoxin (Figure 5), showing that nAChR gene expression led to the respective protein production by GBM cells.

The functions of nAChRs in GBM cell cultures were studied using selective antagonists—α-conotoxins [A10L]PnIA and RgIA, which bind preferably to α7 and α9 nAChRs, respectively [37]. It should be noted that RgIA can also bind to GABA_B_ receptors, which might be expressed by GBM cells. However, the use of RgIA with the respective nAChR agonists choline and acetylcholine allowed for the effects to be monitored selectively. The neurotoxin azemiopsin from *Azemiops feae* viper venom is known to selectively block muscle α1β1δε nAChR [38]. The activity of the receptors was monitored using fluorescent Ca^2+^ imaging. Acetylcholine application to cells evoked an increase in intracellular Ca^2+^, which was detected as a fluorescence intensity increase (Figure 6). If this increase in calcium was depressed by a selective antagonist, it was interpreted as mediated (at least in part) by the respective nAChR subtype. Interestingly, the U87MG cell line showed a significant depression in the acetylcholine-evoked calcium increase by α7 nAChR-targeting [A10L]PnIA only in cells cultured on FBS-supplemented medium (Figure 7A), which is in a good agreement with RT-PCR data showing a decrease in α7 nAChR gene expression under serum-free conditions. However, no significant evidence of functional muscle or α9 nAChR was found in U87MG cells grown on the medium containing either FBS or serum.

The proneural cell line 011 did not show signs of an acetylcholine-evoked calcium increase in either serum-free or serum-supplemented medium (Figure 7B). The other proneural GBM line 019 demonstrated strong evidence of muscle, α7, and α9 nAChRs when cultured on FBS-supplemented medium (Figure 7C). The presence of α7 nAChR was further supported by patch-clamp recordings of whole-cell currents elicited by the selective α7 nAChR agonist in combination with the selective α7 nAChR positive allosteric modulator (Figure 7D). Despite nAChRs being detected by RT-PCR in 019 cells grown on both types of medium, no significant effects of selective inhibitors were detected in calcium-imaging experiments on cells cultured in serum-free conditions (Figure 7C, right panel), suggesting the significant influence of culture conditions on the receptors’ translation or membrane transport. In contrast to proneural GBM lines 011 and 019, mesenchymal lines 022 and 067 both showed strong evidence of α7 nAChR functioning only when cultured on serum-free medium (Figure 7E and Figure 7F, respectively). These results demonstrate the variability of the GBM cell lines in respect to functional nAChR expression.

It was recently reported by Pucci et al. that nicotine and choline increased the proliferation of glioblastoma model cell lines U87MG and GBM5 [19]. In this report, the U87MG line was cultured in FBS-supplemented medium and GBM5 was kept in serum-free medium supplemented with Neurobasal and B27 additives. Interestingly, Neurobasal supplement contains choline at a concentration of 28.6 μM. The concentration of choline in blood serum is estimated to be in the range of 7.1–28.9 μM [30]. Some reports used LC-MS/MS to estimate the choline concentration in blood plasma at a level of 11.4 ng/μL, which corresponds to ~100 μM [29], suggesting that in medium supplemented with 10% FBS, the choline levels can reach concentrations of up to 10 μM. A similar choline concentration was obtained in 1:1 DMEM/F12 with a Neurobasal mixture. Thus, GBM cells under such experimental conditions may be affected by constant muscle, α7, and α9 nAChR activation.

Choline is a metabolic precursor of the major endogenous cholinergic neurotransmitter acetylcholine and plays an important role in nAChR function [39]. It has been shown to act as an agonist of muscle, α7, and α9 nAChRs and as a modulator of α3β4 nAChR [26,40,41]. Additionally, it has recently been shown that choline promotes glioblastoma cell proliferation [19]. In our work, the effects of nAChR agonists on various GBM cell lines were investigated in relation to cell-culturing conditions. To test the possibility of choline influence on the results, serum-free medium was based on the Neuro Brew^TM^ NS-21 supplement (reformulated variant of B27), which does not contain choline.

In the current study, no significant effect of epibatidine (which is an agonist of all nAChR subtypes except α9) on U87MG or patient-derived GBM microsphere cultures was detected in a wide range of concentrations (3 nM–1 μM) (Figure 6). Surprisingly, the application of nAChR antagonists along with epibatidine in some experiments stimulated proliferation: α-conotoxin [A10L]PnIA, which selectively inhibits α7 nAChR, stimulated the proliferation of proneural cell line 011 grown on FBS-supplemented medium (Figure 8A), but no effects of [A10L]PnIA or azemiopsin were detected in serum-free medium (Figure 8B). No effect of nAChR ligands (antagonists or agonists) was detected on the proliferation of proneural cell line 019 (Figure 8C,D) despite this line clearly showing the presence of functional nAChRs in calcium-imaging and patch-clamp experiments. It is widely accepted that nAChR-mediated effects are dependent on protein kinase C activation and subsequent mitogen-activated protein kinase signal cascades [16]. Thus, exact reactions of the GBM to nAChR ligands should depend not only on nAChR subtype surface expression but also on the pre-existence of such mechanisms in GBM cells.

The proliferation of mesenchymal cell lines 022 and 067 and model line U87MG cultured on FBS-supplemented and serum-free media was stimulated by [A10L]PnIA, suggesting the possible role of α7 nAChR, which is in a good agreement with calcium-imaging and RT-PCR results (Figure 8E–J). Azemiopsin, the selective muscle nAChR inhibitor, stimulated proliferation of mesenchymal line 022 and model line U87MG in both types of medium (FBS and serum-free, see Figure 8E,F,I,J), but for mesenchymal line 067 its effect was detected solely in FBS-supplemented medium (Figure 8H).

In contrast to data reported by Pucci et al., the activation of nAChR by epibatidine did not stimulate the proliferation of GBM cells (apart from a subtle effect on the proneural 011 line in serum-free medium, see Figure 8B) [19]. At the same time, the inhibition of α7, α9, and muscle nAChRs stimulated cell proliferation. A similar effect of α7 and α9 nAChR-selective α-conotoxins on glioma cells was previously shown [42]. In our study, the influence of agonists on proliferation was studied in the absence of externally added agonists (Figure 9). Strikingly, on the FBS-supplemented medium, the U87MG model line and proneural GBM patient-derived cultures 011 and 019 demonstrated a significant increase in proliferation in response to α9 nAChR-selective antagonist RgIA (Figure 9A–C). The U87MG and 019 lines also increased proliferation in response to muscle and α7 nAChR-selective antagonists in FBS-supplemented medium (Figure 9A,C). No significant nAChR antagonist-stimulated proliferation was detected on serum-free medium for these lines. Mesenchymal GBM spheroid patient-derived cultures 022 and 067 did not increase proliferation in response to nAChR antagonists on either FBS-supplemented or serum-free medium (Figure 9D,E). In these experiments, the response to antagonists was higher in cells grown in FBS-supplemented medium. Two explanations for these results have been proposed: (i) nAChR agonist choline normally contained in blood serum increases cell surface expression of the respective nAChRs, which partially correlates with RT-PCR results (Figure 4), e.g., line 011 showed an increased expression of the α9 nAChR gene under FBS conditions in comparison with serum-free medium. In the same line, proliferation increased in the FBS presence in response to α9 nAChR-selective antagonist RgIA. (ii) Cell cultivation in serum-supplemented medium leads to an increase in cell differentiation due to a lower concentration of growth factors in serum-supplemented medium than in serum-free medium supplemented with EGF and bFGF [22]. Cells differentiation leads to more stable proliferation rates, i.e., a lack of response to external stimuli (nAChR antagonists).

Since nAChR inhibitor-provoked proliferation tends to emerge (although not exclusively) in FBS-supplemented medium, which by design contains choline as a component of blood serum, the possible role of choline was studied via external addition n varying concentrations to the serum-free medium (Figure 10). No strong evidence of choline provoking cell proliferation was detected on serum-free medium for the U87MG, 011, 019, or 022 line (Figure 10A–D). There was some indication of 100 nM of choline stimulating proliferation of the 067 mesenchymal line (Figure 10E). However, similar to the results of the epibatidine experiment (Figure 8), the addition of selective antagonists along with choline provoked proliferation of the 011, 019, and 067 lines (Figure 10B,C,E). Note that no antagonist-stimulated proliferation was detected on the same serum-free medium when no choline was added (Figure 9), suggesting that some basal nAChR activation by choline is needed for nAChR antagonist-stimulated proliferation.

The serum-free medium used in current study contained EGF and bFGF. EGF has previously been shown to support growth and tumor spread [43]. In particular, EGF is secreted by tumor-associated macrophages and microglia. EGF secretion plays a role in the mechanisms of healing and regeneration of brain tissues, determining the migration of microglia to GBM lesions [44]. EGF influences glial-mesenchymal transition and promotes the microevolution of GBM malignancy and enhances the invasive potential of GBM cells and their ability to penetrate healthy tissues [45]. Thus, the conditions for growing cell cultures determine the heterogeneity of the cell population and their microevolution even in the absence of an active microenvironment (e.g., tumor-associated macrophages). To account for the possible influence of growth factor combinations on nAChR antagonist-stimulated proliferation, the 067 line was grown on NS-21-supplemented serum-free medium, bFGF-supplemented serum-free medium (NS-21/bFGF), or plain serum-free DMEM/F12 (Figure 9). Notably, on medium supplemented solely with NS-21, no significant effects of epibatidine or nAChR antagonists in the presence of epibatidine were detected (Figure 11A). Cell proliferation in the presence of bFGF was stimulated by α7 nAChR antagonist [A10L]PnIA (in mixture with epibatidine), but no effects of epibatidine itself were detected (Figure 11B). Interestingly, on plain serum-free DMEM/F12, epibatidine significantly reduced cell proliferation, and its effect was suppressed by muscle nAChR-selective inhibitor azemiopsin and by α7 antagonist [A10L]PnIA (Figure 11C).

Somewhat unexpectedly, azemiopsin significantly stimulated proliferation of 067 cells grown on NS-21-supplemented serum-free medium without EGF or bFGF (Figure 11D). The addition of bFGF to the medium diminished azemiopsin-stimulated proliferation and promoted the effects of [A10L]PnIA and RgIA, which might indicate that bFGF enhances α7 and α9 nAChR expression or function but inhibits the expression of muscle nAChR (Figure 11E). No pro-proliferative effect of antagonists was detected on plain DMEM/F12 serum-free medium (Figure 11F), suggesting that despite the functional expression of muscle and α7 nAChRs (compare Figure 11C,F), antagonist-stimulated effects are dependent on the activation of the receptor by an agonist. NS-21, on which the Neuro Brew^TM^ used in the current study is based, contains ethanolamine [46]. To our knowledge, ethanolamine does not activate human nAChRs but is a predecessor of the choline and acetylcholine in the biosynthetic pathways, which might explain the significant stimulation of cell growth by α7 and α9 nAChR antagonists on serum-free medium containing Neuro Brew^TM^ (Figure 11). Additionally, the distinct anti-proliferative effect of nAChR agonist epibatidine on plain medium might be explained by the increase in intracellular Ca^2+^ levels above a critical toxic threshold due to chronic nAChR activation. This effect might be suppressed by nAChR inhibition. However, no cytotoxic effect of nAChR activation has been shown in medium-containing growth factors. In this light, it should be mentioned that both EGF and bFGF show neuroprotective effects against calcium-induced excitotoxicity in vitro [47].

## 4. Conclusions

In our paper, which analyzes GBM cell lines, the relationship between culturing condition and the expression of mRNA for nAChR subunits and of functional nAChR receptors is described for the first time using nAChR subtype-selective neurotoxins as research instruments.

We found highly variable consequences of nAChR activation and inhibition on patient-derived GBM spheroid cultures and a U87MG model line. Not only did nAChR subunit gene expression vary between different patients but also actual receptor function and cell proliferation depended on the nature of nAChR ligands. In this study, it was shown that the α1, α7, and α9 subunit-selective antagonists enhanced cell proliferation of the patient-derived glioblastoma cell lines under various culturing conditions. The results reported here might be helpful in subsequent study design in regard to cell-culturing conditions and the interpretation of nAChR ligands’ effects.

## 5. Materials and Methods

### 5.1. Media for the Cultivation of GBM Cells

Medium I (FBS-supplemented medium): DMEM/F12 (Sigma–Aldrich, Taufkirchen, Germany) containing 2 mM glutamine, 10% fetal bovine serum (FBS) (Gibco, Thermo Fisher Scientific, Waltham, MA, USA), 1% penicillin–streptomycin solution (Sigma–Aldrich, Germany).

Medium II (serum-free medium): DMEM/F12 (Sigma–Aldrich) containing 2 mM glutamine, 2% NS-21 (MACS NeuroBrew-21 supplement (Miltenyi Biotec, Waltham, MA, USA)), 20 ng mL^−1^ basic fibroblast growth factor (bFGF; Sigma–Aldrich), 20 ng mL^−1^ epidermal growth factor (EGF; Sigma–Aldrich), 1% penicillin–streptomycin solution (Sigma–Aldrich).

Medium III: DMEM/F12 (Sigma–Aldrich) containing 2 mM glutamine, 2% NS-21 (MACS NeuroBrew-21 supplement (Miltenyi Biotec)), 20 ng ml^−1^ basic fibroblast growth factor (bFGF; Sigma–Aldrich), 1% penicillin–streptomycin solution (Sigma–Aldrich).

Medium IV: DMEM/F12 (Sigma–Aldrich) containing 2 mM glutamine, 2% NS-21 (MACS NeuroBrew-21 supplement (Miltenyi Biotec)), 1% penicillin–streptomycin solution (Sigma–Aldrich).

Medium V: DMEM/F12 (Sigma–Aldrich) containing 2 mM glutamine, 2%, 1% penicillin–streptomycin solution (Sigma–Aldrich).

### 5.2. Cell Culture

All cells were cultured at 37 °C in a humidified atmosphere with 5% CO_2_. bFGF and EGF were added twice weekly, and the culture medium was changed every 5–10 days. To attach the cells to the glass or plastic, the surface was preincubated overnight at room temperature with a solution of laminin in PBS (1:200). Patient-derived culture microspheres were dissociated using StemPro Accutase (Thermo Fisher Scientific, Waltham, MA, USA). U87MG cells were dissociated using Trypsin-Versene solution (PanEco, Moscow, Russia).

### 5.3. RNA Isolation and RT-qPCR

RNA was isolated using the ExtractRNA kit (Evrogen, Moscow, Russia). RNA concentration was determined using a Nanodrop One C spectrophotometer (Thermo Fisher Scientific). cDNA was synthesized using the MMLV reverse transcription kit (Evrogen) according to the manufacturer’s protocol. qPCR was performed on a LightCycler 96 (Roche, Basel, Switzerland) with qPCRmix-HS SYBR reagent (Evrogen). The cycling conditions were 95 °C for 150 s, followed by 45 cycles of 95 °C for 20 s, 60 °C for 20 s, and 72 °C for 20 s. Data were collected using LightCycler software (version 4.1). The 18S RNA was used as an intermediate control. The primer sequences are provided in Appendix A.

### 5.4. Single-Cell Calcium Imaging

Patient-derived cultures of GBM cells, as well as U87MG cells, were grown in 96-well plates in media I and II at 37 ° C, 5% CO_2_ atmosphere, 100% humidity. Before calcium imaging began, the growing medium was replaced with extracellular buffer (140 mM NaCl, 2 mM CaCl_2_, 2.8 mM KCl, 4 mM MgCl_2_, 20 mM HEPES, 10 mM glucose, pH 7.4), and then each well was loaded with a cell-permeant 2,5 μM Fluo-4AM solution (ex/em = 494/506 nm; Thermo Fisher Scientific) for 40 min. The fluorescent dye solution was further removed, and cells were kept in an extracellular buffer for 1 h. Ca^2+^ dynamics were recorded using an Olympus IX71 epifluorescent microscope with an appropriate filter combination and a CAM-XM10 cooled CCD camera (Olympus, Tokyo, Japan). The cells were exposed to 10 μM of nAChR agonist acetylcholine iodide (Sigma-Aldrich), 1 μM of antagonists Azemiopsin, and [A10L]PnIA or RgIA (Syneuro, Moscow, Russia), and changes in the fluorescence of calcium indicator Fluo-4 were recorded for each cell independently. Videos were recorded and processed using CellA imaging software version 3.1 build 1274 (Olympus Soft Imaging Solutions GmbH, Münster, Germany). Data analysis was performed using open-source ImageJ Fiji software (version 1.54f), where changes in fluorescent intensity per cell before and after the nAChR ligand exposure were calculated. The response of at least 5 cells was measured.

### 5.5. Patch-Clamp

Cells were grown on laminin-covered glass placed on 24-well plates under the same conditions described in Section 2.4. To conduct the electrophysiological recording of the α7 nAChR-mediated macroscopic currents, a whole-cell patch clamp was set up as follows. Cells attached to the glass were transferred to a bath filled with extracellular electrophysiology solution (140 mM NaCl, 2 mM CaCl_2_, 2.8 mM KCl, 4 mM MgCl_2_, 20 mM HEPES, 10 mM glucose, pH 7.4). A microelectrode pipette was filled with intracellular buffer solution (140 mm CsCl, 6 mm CaCl_2_, 2 mm MgCl_2_, 2 mm MgATP, 0.4 mm NaGTP, 10 mm HEPES/CsOH, 20 mm BAPTA/KOH). A microelectrode was attached to the cell membrane until a resistance of at least 1 GOhm was reached. After establishing the gigaseal, the cell membrane was ruptured using suction and the recordings were created. A typical experiment consisted of 1 s cell wash with control extracellular buffer, then the bath solution was changed to an extracellular solution with 1, 10, or 100 μM of α7 nAChR-selective agonist PNU-282987 (Tocris, Bristol, UK) supplemented with 10 μM of α7 nAChR-selective positive allosteric modulator PNU-120596 (Tocris) for 5 s followed by 5 s wash-out with control extracellular buffer.

### 5.6. Fluorescent α-Bungarotoxin Binding Assay

GBM cells were grown on laminin-covered glass placed on 24-well plates under the same conditions described in Section 2.4. To assess the cell expression of α7, α9, and muscle-type nAChRs, the cells were fixed with 4% PFA and then stained with Alexa-Fluor 555-conjugated α-bungarotoxin (50 nM, α-Bgt-Alexa 555, Thermo Fisher Scientific) overnight at 37 °C. The cells were washed extensively with extracellular buffer (140 mM NaCl, 2 mM CaCl2, 2.8 mM KCl, 4 mM MgCl2, 20 mM HEPES, 10 mM glucose, pH 7.4) to remove any unbound α-bungarotoxin. Controls were run simultaneously with 1 μM of unlabeled d-tubocurarine (Tocris). Twelve-bit digital images were obtained using a DuoScanMeta LSM510 laser scanning microscope (Carl Zeiss, Weimar, Germany) equipped with a Plan-Apochromat 63×/1.40 objective (numerical aperture). Image acquisition parameters were as follows: for green fluorescence—excitation at 488 nm and emission at 505–550 nm, for red fluorescence—excitation at 561 nm and emission at 575 nm. Pictures were processed with open-source ImageJ Fiji software version 1.54f.

### 5.7. Cell Proliferation Assay

Cells were plated on a 96-well plate at a density of 6000 cells per well in 150 μL medium. The number of cells was assessed using AlamarBlue reagent (Thermo Fisher Scientific). The fluorescence was measured using a Fusion α-FP HT Universal Microplate Analyzer (PerkinElmer, Waltham, MA, USA) with an excitation filter for 535 nm and an emission filter for 620 nm. The measurements were taken on day 5.

### 5.8. Data and Statistical Analysis

The non-parametric Kruskal–Wallis test and Dunn’s post-hoc test were used to study the significance of the effects in the calcium-imaging assay due to the obvious deviation of response amplitudes from normal distribution. One-way or two-way ANOVA with Tukey’s or Dunnett’s post-hoc test was used to analyze cell viability assays.

## Figures and Tables

**Figure 1 toxins-16-00080-f001:**
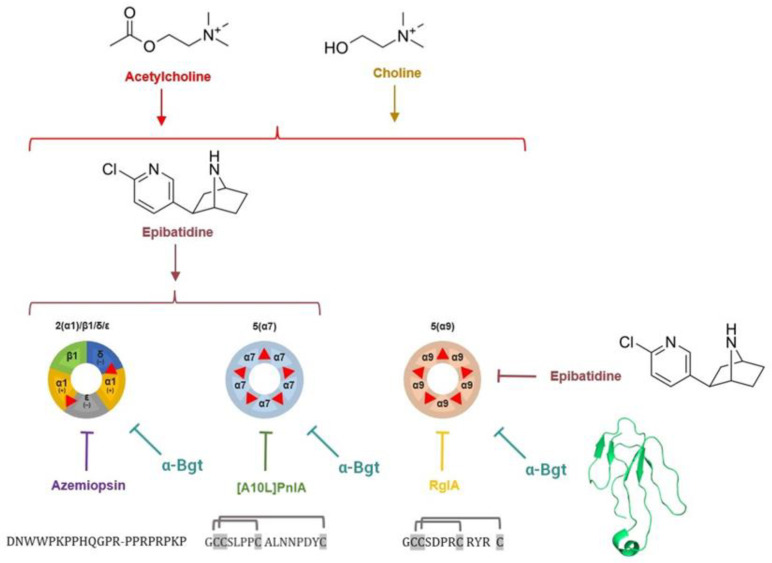
Subtypes of nAChR studied in this work: muscle-type heteropentameric α1β1δε receptor and neuronal α7 and α9 homopentameric receptors. Orthosteric binding sites for agonists and competitive antagonists are shown as red triangles. Orthosteric agonists choline, acetylcholine, and epibatidine are shown as chemical formulae; peptide neurotoxins azemiopsin, [A10L]PnIA, and RgIA are shown by their respective sequences (braces denote disulfide bonds); and small protein α-bungarotoxin is shown by ribbon representation of spatial structure. Azemiopsin is the neurotoxic peptide that has been identified in *Azemiops feae* viper venom. [A10L]PnIA is the modified α-conotoxin that has been identified in the venom of marine snail *Conus pennaceus*, and α-conotoxin RgIA has been identified in *Conus regius* venom. The peptide toxins used in this study were chemically synthesized in house and described elsewhere. Arrows denote activation of the receptor, T-shaped arrows denote receptor inhibition.

**Figure 2 toxins-16-00080-f002:**
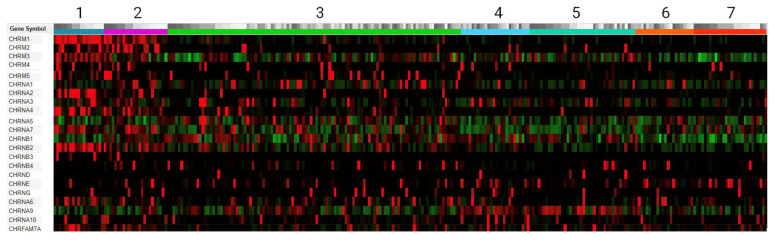
Differential expression of acetylcholine receptor genes (RNA-Seq mRNA data from Ivy GAP). Gene expression values are sorted by regions and varieties of GBM from which samples were taken: (1) leading-edge samples (dark blue); (2) infiltrating tumor samples (magenta); (3) cellular tumor samples (green); (4) peri-necrotic-zone samples (light blue); (5) pseudopalisading cells around necrosis zone samples (Caribbean green); (6) hyperplastic blood vessels in cellular tumor samples (orange); (7) microvascular proliferation samples (red). The expression level is compared relative to the z-score value. A z-score value other than zero indicates the standard deviation from the average expression value of this gene. Therefore, a positive and a negative z-score indicates increased and decreased gene expression, respectively. On the given heatmap, a high z-score value is indicated in red, a z-score value below zero is depicted in green, and a z-score value equal to or close to zero is represented by black.

**Figure 3 toxins-16-00080-f003:**
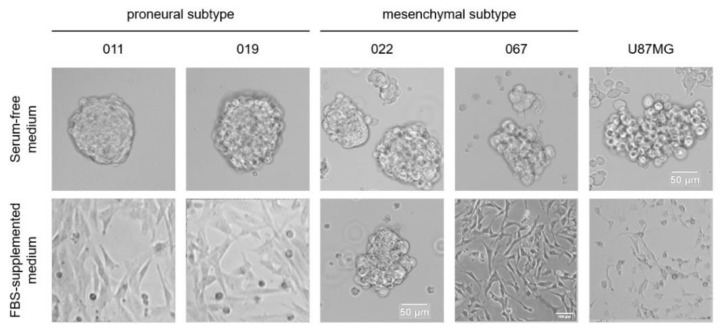
Influence of cultivation conditions of patient-derived GBM cell lines and U87MG cell line in vitro on their proliferation. GBM cell lines cultured in serum-free medium grew as microspheres. In serum-supplemented medium, almost all cell lines proliferated as an attached monolayer and did not form microspheres.

**Figure 4 toxins-16-00080-f004:**
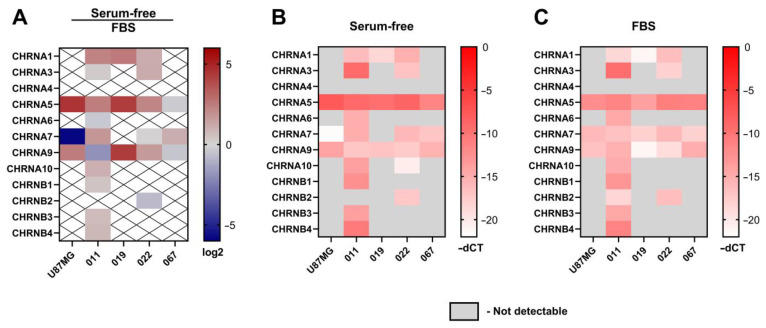
Comparative qPCR analysis of nAChR subunits genes expressed in patient-derived GBM lines and model line U87MG. (**A**) Relative gene expression was calculated as base 2 logarithm intensity values (serum-free samples over FBS samples) and represented as a heatmap. Red denotes a gene expression increase; blue is genes that decreased the expression level in serum-free compared to FBS-supplemented medium. (**B**) nAChR gene expression in serum-free medium (medium II, containing bFGF and EGF growth factors and NS-21 neuronal vitality supplementation) in terms of dCT values. The threshold cycle values of nAChR subunit genes were corrected to the 18S level in the corresponding cell lines cultured in serum-free conditions. Red denotes higher expression. (**C**) nAChR gene expression in FBS-supplemented medium in terms of dCT. The threshold cycle values of nAChR subunit genes were corrected to the 18S level in the corresponding cell lines cultured in FBS-supplemented medium. Red denotes higher expression.

**Figure 5 toxins-16-00080-f005:**
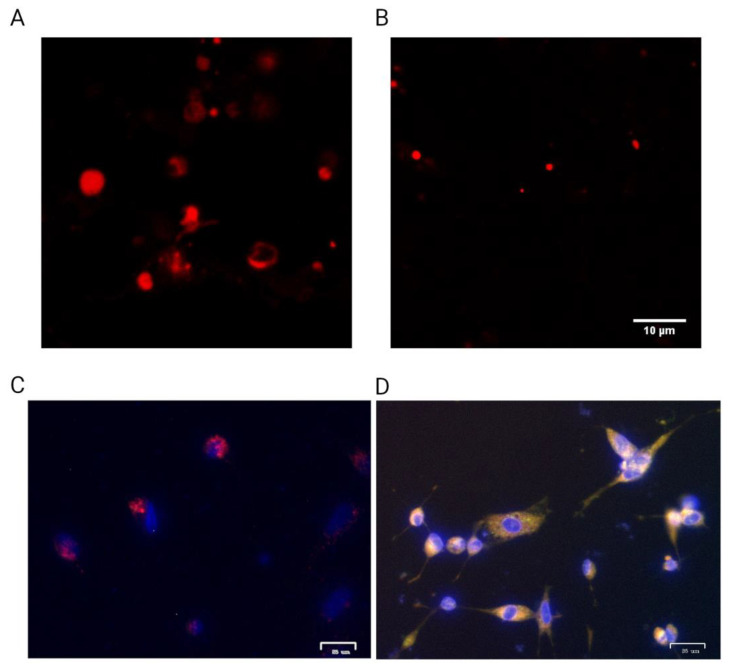
Visualization of nAChRs via confocal laser scanning microscopy. (**A**) Mesenchymal GBM cell line 067 incubated with α-Bgt-Alexa 555; (**B**) the same GBM cells co-incubated with α-Bgt-Alexa 555 and an excess amount of d-tubocurarine (d-TC); (**C**) proneural GBM line 019 cells co-stained with α-Bgt-Alexa 555 and DAPI; (**D**) proneural GBM line 019 cells co-stained with DAPI and membrane fluorescent Dil lipophilic membrane stain.

**Figure 6 toxins-16-00080-f006:**
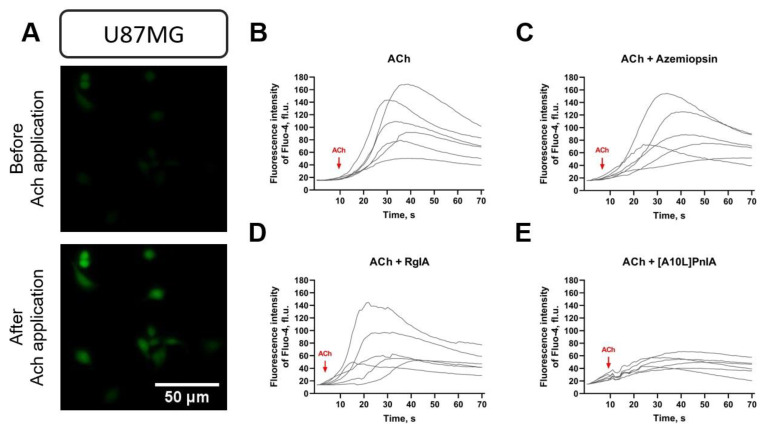
Registration of transient fluorescent signals from the cytoplasmic calcium indicator Fluo-4 AM in U87MG cells grown on FBS-supplemented medium and exposed to nAChR ligands. (**A**) The microphotographs show the change in indicator fluorescence intensity before and after application of acetylcholine (ACh). (**B**) Application of ACh caused an increase in intracellular Ca^2+^ levels, which registered as an increase in fluorescence intensity (fluorescence units, fl.u.). (**C**,**D**) No significant effect of the inhibitors azemiopsin or RgIA on the ACh-stimulated change in Ca^2+^ levels was detected, indicating the non-involvement of the muscle-type and α9 nAChRs. (**E**) The α7 nAChR antagonist [A10L]PnIA prevented an ACh-stimulated increase in intracellular Ca^2+^ levels. Each curve shows the change in intracellular Ca^2+^ concentration in a single cell. The arrow indicates the time of agonist application.

**Figure 7 toxins-16-00080-f007:**
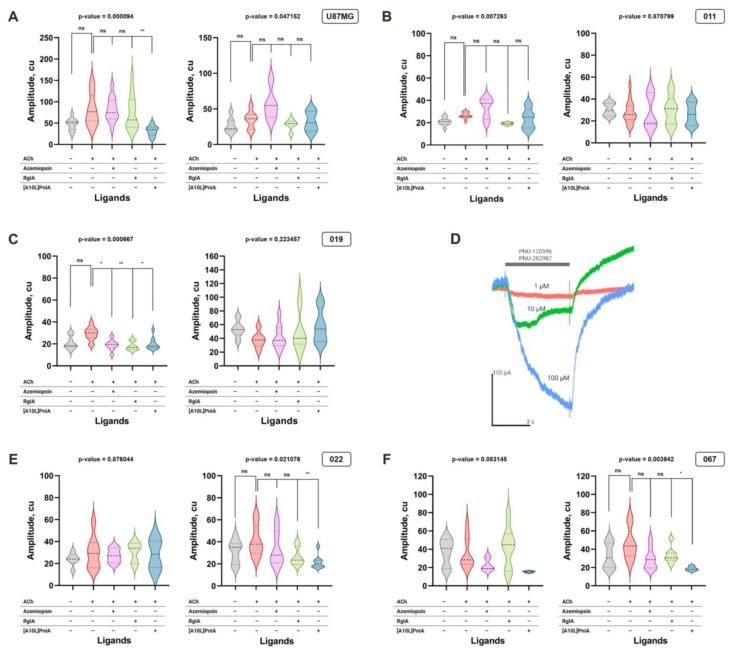
Effects of nAChR ligands on the intracellular Ca^2+^ concentration in GBM. Cells were loaded with Fluo-4 fluorescent calcium indicator. The left diagram of each panel shows the acetylcholine-evoked Ca^2+^ increase in cells grown on FBS-supplemented medium; the right diagram of the panel demonstrates the responses of the cells in serum-free (NS-21/EGF/bFGF-supplemented) medium. The data are presented for U87MG model cell line (**A**) as well as the GBM cell lines 011 (**B**), 019 (**C**), 022 (**E**), and 067 (**F**). The transient signal amplitude of the calcium indicator is measured in conventional units, c.u. For the violin plots, the black horizontal dotted lines display the mean as well as the 25th and 75th percentiles. Panel (**D**) shows the concentration-dependent ion currents in the whole-cell patch-clamp experiment for cell culture 019, grown in FBS-supplemented medium, in response to the mixture of α7 nAChR-selective agonist PNU 282987 and positive modulator PNU 120596. *p*-values were computed using the Kruskal–Wallis test and pairwise Dunn’s test. Asterisks indicate significant differences at * *p* < 0.05 and ** *p* < 0.01; ns, *p* > 0.05.

**Figure 8 toxins-16-00080-f008:**
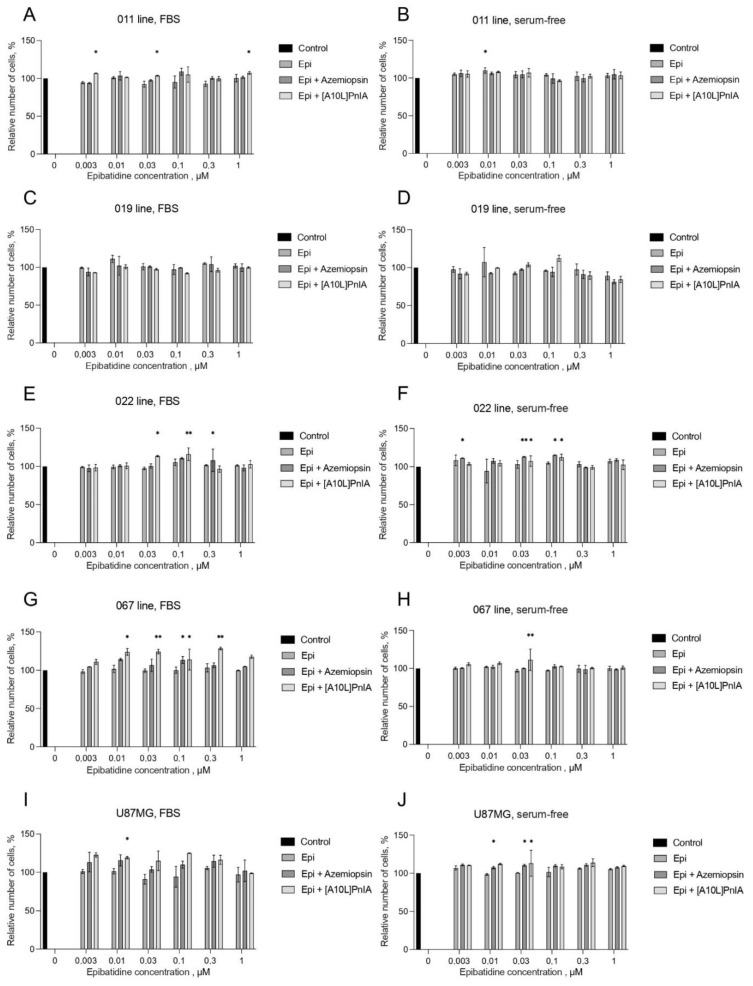
Influence of nAChR ligands on cell proliferation in FBS-containing (**A**,**C**,**E**,**G**,**I**) and serum-free (**B**,**D**,**F**,**H**,**J**) medium assessed by AlamarBlue assay. Values were normalized to control without any ligand. The data are presented for GBM cell lines 011 (**A**,**B**), 019 (**C**,**D**), 022 (**E**,**F**), and 067 (**G**,**H**), as well as for U87MG (**I**,**J**). The concentration of azemiopsin and [A10L]PnIA was 1 μM. Data are represented as the mean ± SD; for all experiments *n* = 5. *p*-values were computed using one-way ANOVA followed by Dunnett’s test. Asterisks indicate significant differences at * *p* < 0.05 and ** *p* < 0.01.

**Figure 9 toxins-16-00080-f009:**
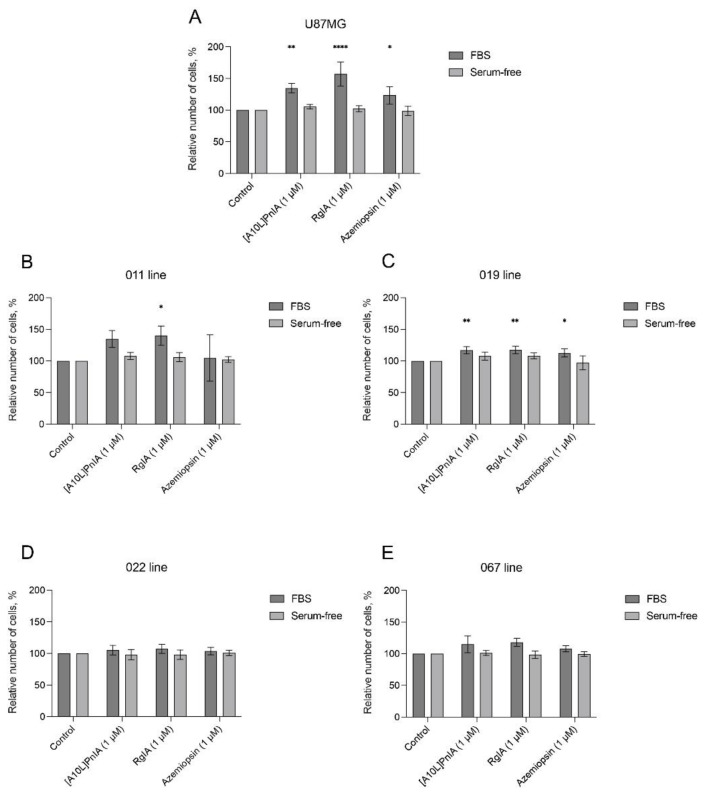
The effects of nAChR antagonists on the proliferation of GBM cells under different cultivation conditions assessed by AlamarBlue assay. Values were normalized to control without ligands. The data are presented for the U87MG model cell line (**A**) as well as GBM cell lines 011 (**B**), 019 (**C**), 022 (**D**), and 067 (**E**). The concentration of azemiopsin, RgIA, and [A10L]PnIA was 1 μM. Data are represented as the mean ± SD; for all experiments *n* = 5. *p*-values were computed using one-way ANOVA followed by Dunnett’s test. Asterisks indicate significant differences at * *p* < 0.05, ** *p* < 0.01 and **** *p* ≤ 0.0001.

**Figure 10 toxins-16-00080-f010:**
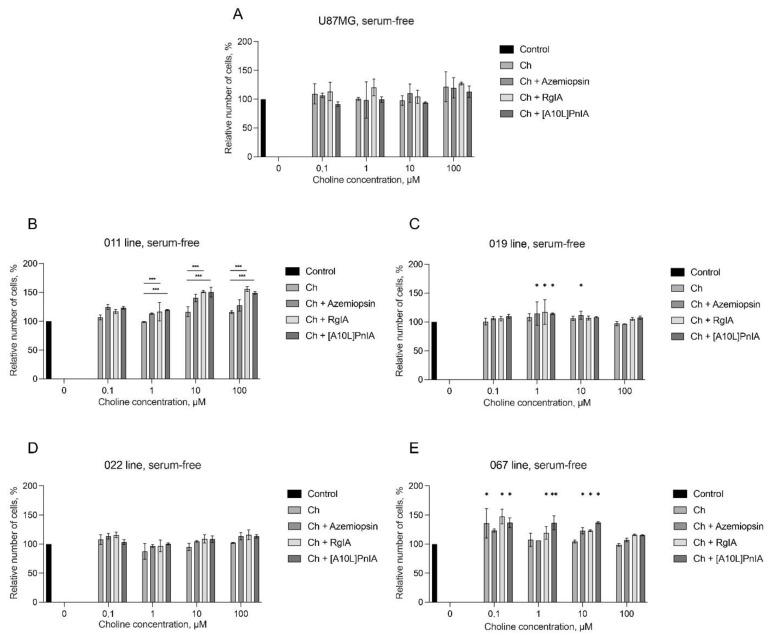
The effect of nAChR stimulation with choline alone and in combination with different antagonists on the proliferation of GBM cells in serum-free medium. AlamarBlue assay was used. Values were normalized to control without ligands. The data are presented for the U87MG model cell line (**A**) as well as GBM cell lines 011 (**B**), 019 (**C**), 022 (**D**), and 067 (**E**). The concentration of epibatidine, azemiopsin, RgIA, and [A10L]PnIA was 1 μM. Data are represented as the mean ± SD; for all experiments *n* = 5. *p*-values were computed using one-way ANOVA followed by Dunnett’s and Tukey’s test. Asterisks indicate significant differences at * *p* < 0.05, ** *p* < 0.01, and *** *p* ≤ 0.001.

**Figure 11 toxins-16-00080-f011:**
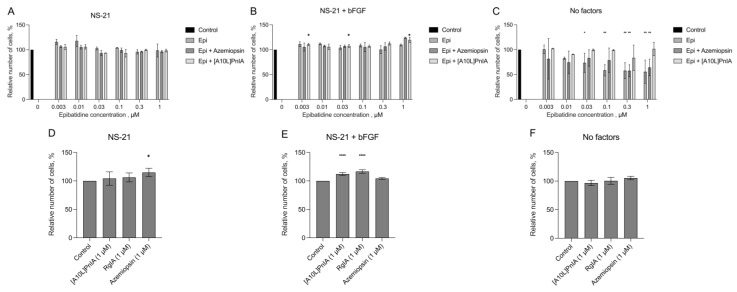
The effects of nAChR ligands on the proliferation of the 067 GBM cell line under various cultivation conditions. AlamarBlue assay was used. Values were normalized to control without ligands. The data are presented for NS-21-supplemented serum-free medium without the addition of EGF or bFGF (**A**,**D**), NS-21-supplemented medium with bFGF (**B**,**E**), and plain DMEM/F12 medium (**C**,**F**). Data are represented as the mean ± SD; for all experiments *n* = 5. *p*-values were computed using one-way ANOVA followed by Dunnett’s test. Asterisks indicate the significant differences at * *p* < 0.05, ** *p* < 0.01, and **** *p* ≤ 0.0001.

## Data Availability

All data reported in the manuscript are available from the corresponding author on demand.

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
