# Peer review of "Subtype-Selective Peptide and Protein Neurotoxic Inhibitors of Nicotinic Acetylcholine Receptors Enhance Proliferation of Patient-Derived Glioblastoma Cell Lines"

_toxins, 2024, doi:10.3390/toxins16020080_

Round 1

Reviewer 1 Report

Comments and Suggestions for Authors

In this manuscript, the authors investigated the effect of culture conditions of patient-derived glioblastoma cells on (i) the expression of different nAChR subtypes and (ii) the functional activity of these receptors using specific antagonists.

They found that activation and inhibition of patient-derived GBM cultures can have a very different impact on nAChR subunit gene expression and cell proliferation depending on the ligand tested.

My main concern is that this work is quite descriptive and no precise explanation has been provided to explain the huge variation in the effect observed on the different GBM cell lines depending on the antagonist tested, their concentration or the presence or absence of agonists. Furthermore, the proliferative effects of nAChR antagonists are in contradiction with many previous publications reporting an anti-proliferative effect of this type of ligands or siRNA in many different cancer cells.

In addition, certain discrepancies appeared in the results obtained in the different experiments making certain conclusions confusing. For example, in Figure 8, an effect of azemiopsin on the 067 cell line is reported while no alpha1 receptor was detected in the qPCR in Figure 4. How can this result be explained? In all these experiments, the specificity of the reported effect must be validated with other ligands specific for the different nAChR subtypes, such as for example PNU282987 or dimethyl-4-phenylpiperazinium iodide for the selective activation of a7 or a1 or MLA and d-tubocurarine for selective inhibition of a7 or a1.

How the authors explain the difference in effect reported at different concentrations of epibatidine. For example, potentiation of PnIA was only observed at 0.01 µM Epi on U97MG and not at 0.003 or 0.03???

In all cell proliferation assays, the n value for each point is not reported. It is particularly important to evaluate the strength of the statistical analysis, particularly because of the small effect generally observed in this test (10-20% increase in proliferation in general).

Minor points :

Line 69 : MLA is not mecamylamine but methyllycaconitine and it is a  alpha7-selective antagonist

Fig. 1 : The sequence reported corresponds to PnIA and not PnIA-A10L

Why RgIA is tested alone in Fig 9 and not studied as other antagonists with Epibatidine in Fig 8 ??

The use of RgIA can be problematic due to its reported effect on GBM proliferation via GABA receptor activation

Line 185 : « The fluorescent ligand binding can be attributed to nAChR of either α7, α9 or muscle-type ». According to the Fig. 4, the a1 subunit is not present in this cell line (067), so the labeling cannot be associated with the muscle-type receptor ?

How explain the large heterogenity in the Ca response in different cells reported in fig.6

The concentrations of antagonists are not indicated in fig. 8

Author Response

In this manuscript, the authors investigated the effect of culture conditions of patient-derived glioblastoma cells on (i) the expression of different nAChR subtypes and (ii) the functional activity of these receptors using specific antagonists.

They found that activation and inhibition of patient-derived GBM cultures can have a very different impact on nAChR subunit gene expression and cell proliferation depending on the ligand tested.

My main concern is that this work is quite descriptive and no precise explanation has been provided to explain the huge variation in the effect observed on the different GBM cell lines depending on the antagonist tested, their concentration or the presence or absence of agonists. Furthermore, the proliferative effects of nAChR antagonists are in contradiction with many previous publications reporting an anti-proliferative effect of this type of ligands or siRNA in many different cancer cells.

We thank the Reviewer for thorough analysis of the text. The work presented in current manuscript was designed as a starting point of our project aiming at investigation of various toxins action on primary GBM lines obtained from real-world patients. Indeed, this first paper dedicated to this subject looks very much like descriptive study and we tried to avoid overly suggestive interpretation of the results.

It should be noted that barely see any influence of nAChR ligands on our cell lines viability despite clearly demonstrated functional activity of the receptors on these cells. Strong effects are observed under certain specific conditions only (e.g. Figures 10 and 11 where serum-free or even growth factor-free conditions were explored). Our main idea is to illustrate the great variety of the toxins (and other nAChR ligands) effects. This may help our colleagues to avoid undesirable situations in experimental design.

In addition, certain discrepancies appeared in the results obtained in the different experiments making certain conclusions confusing. For example, in Figure 8, an effect of azemiopsin on the 067 cell line is reported while no alpha1 receptor was detected in the qPCR in Figure 4. How can this result be explained?

Indeed, bulk nAChR gene expression analysis did not reveal significant alpha1 (muscle nAChR) levels in both FBS and serum-free media. Moreover, no effects of azemiopsin on calcium transients were detected in fluorescent assay on the line 067. Azemiopsin is very selective muscle nAChR antagonist, so we tend to excude the possibility of some other receptor involvement in the effects observed. We see two possibilities for azemiopsin to manifest its effects in cell viability assay: i) long term application of nAChR ligands, which was performed in case of alamar blue tests but not qPCR or calcium imaging, promotes nAChR expression. Such an effect is well-known for nicotine that promotes nAChR expression (https://www.ncbi.nlm.nih.gov/pmc/articles/PMC3829154/); ii) small and concentration-independent effect in the presence of epibatidine only on FBS-supplemented medium could probably be considered an outlier. In such tests with multiple comparisons no statistical test guarantees against false positive results and all such results should be interpreted in comparison to other experiments (such as qPCR and calcium imaging). The manuscript has been ammended to address this concern.

In all these experiments, the specificity of the reported effect must be validated with other ligands specific for the different nAChR subtypes, such as for example PNU282987 or dimethyl-4-phenylpiperazinium iodide for the selective activation of a7 or a1 or MLA and d-tubocurarine for selective inhibition of a7 or a1.

In our work we used decently good and selective ligands, such as [L10A]PnIA, RgIA and azemiopsin. But we agree with the reviewer that wider panel of ligands can add credibility to the data obtained in this study. We add supplementary data on calcium fluorescence assay of selective α7 nAChR ligands activity on 022 GBM cell line grown on FBS-supplemented medium (S2). We also performed Alamar Blue assay of 022 cells viability under the treatment with α7 nAChR-selective agonist PNU-282987 (concentrations varying from 20 nM to 6000 nM) and nAChR antagonist α-cobratoxin at 1000 nM (S3).

We’ll continue to study all described cell lines using other methods and wider panel of ligands. These results would be published elsewhere.

How the authors explain the difference in effect reported at different concentrations of epibatidine. For example, potentiation of PnIA was only observed at 0.01 µM Epi on U97MG and not at 0.003 or 0.03???

As said previously, we tend to interpret such effects as a statistical outliers. We want to stress out that the main result in this experiment is the absence of GBM cells growth inhibition (somewhat in contrast to previously published data). This may help our colleagues to critically revise experimental conditions used to detect such antiproliferative effects (e.g. medium and growth factors). In that sense, such “negative” results could be valuable to the scientific community.

In all cell proliferation assays, the n value for each point is not reported. It is particularly important to evaluate the strength of the statistical analysis, particularly because of the small effect generally observed in this test (10-20% increase in proliferation in general).

We agree and have updated the descriptions of figures 8-10. The weakness of the observed effects is actually interesting point itself. Why neither nAChR agonists that explicitly evoke calcium rise in GBM cells, nor toxins that clearly inhibit nAChRs, did not unequivocally influence cell viability? We’ll study these effects in more details using proteomics approach and publish results in our next papers.

Minor points:

Line 69: MLA is not mecamylamine but methyllycaconitine and it is an alpha7-selective antagonist

We thank the reviewer for pointing this out. We have fixed the error.

Fig. 1: The sequence reported corresponds to PnIA and not PnIA-A10L

We have modified the Figure 1 and have changed the sequence of [A10L]PnIA.

Why RgIA is tested alone in Fig 9 and not studied as other antagonists with Epibatidine in Fig 8??

Epibatidine activates all nAChR subtypes except α9α10. Epibatidine acts as an antagonist of α9α10 nAChR. RgIA also inhibits this receptor subtype, therefore, we did not use these two ligands together.

The use of RgIA can be problematic due to its reported effect on GBM proliferation via GABA receptor activation

We agree that such a problem exists. The respective note has been added to the discussion section. However, in our view, RgIA is still a gold standard in alpha9 nAChR-related research because it was the most extensively studied among conotoxins (that is why we know about its off-target effects on GABAB). On the other hand, we tested the activity of RgIA in response to AChR-specific agonists choline and acetylcholine which to some extent helps to distinguish effects of nAChR and GABAB receptors.

We’ll add more selective Rg2A, oligoarginines and GeXIVA analogues in our future works to study the alpha9 role in more details.

Line 185: «The fluorescent ligand binding can be attributed to nAChR of either α7, α9 or muscle-type». According to the Fig. 4, the a1 subunit is not present in this cell line (067), so the labeling cannot be associated with the muscle-type receptor?

We have revised this sentence. The new sentence reads as follows “Since the expression of α1 nAChR subunit was not detected on 067 cell line via RT-PCR, the fluorescent ligand binding can be attributed to nAChR of either α7 or α9”.

How explain the large heterogenity in the Ca response in different cells reported in fig.6

As noted in the caption to the Figure 6, each individual line shows the change in intracellular Ca2+ concentration in a single cell. The glioblastoma cell cultures studied in this research are heterogeneous, consequently, differences in the amplitude of recorded signals between cells within the same cell culture were observed. For further statistical analysis we used the average amplitude of transient signals.

The concentrations of antagonists are not indicated in fig. 8

We agree and have updated the descriptions of figures 8-10.

Reviewer 2 Report

Comments and Suggestions for Authors

Reference:  “Subtype-selective peptide and protein neurotoxic inhibitors of nicotinic acetylcholine receptors enhance proliferation of patient-derived glioblastoma cell lines”. Submitted to TOXINS, December, 2023.

General comments: Glioblastomas represent a type of cancer classified as Big Killers, due to their high capacity to kill affected patients. Any novelty that brings serious knowledge in this area, and that can serve to mitigate and alleviate the suffering of affected patients deserves respect and serious judgment. In this manuscript the authors are studying muscarinic acetylcholine receptors and the role of neurotoxins in these receptors presented in Glioblastoma multiforme lineage (U87MG GBM cell line) and primary cultivated cells obtained from patients with these tumors. Throughout the text, the authors report their findings, which show neutroxins that present activity on the subtypes of nicotinic acetylcholine receptors α1, α7 and α9, when exposed to Glioblastoma U87MG GBM cell lines, as well as cells obtained from patients with glioblastoma , and treated with nicotinic agonists or antagonists, showed proliferative activity on these cells under specific culture cell conditions. The data strengthen the use of primary cultures as models to study this type of tumor, as well as opening the way for learning about the cell biology of glioblastomas.  After carefully reading the text, it is my opinion that this is a subject that falls within the scope of TOXINS, a subject of interest in public health, and finally a research that describes the participation of cellular acetylcholine receptors influencing tumor cell growth. Attached are some suggestions that could make a modified version of the text more complete and attractive.

Specific Comments:

1- In the line 25 the authors wrote…. …shown that nAChRs subtype-selective antagonists increase proliferation … In my opinion it is better to write the name of antagonists, as there are several types that could act in different ways.  

2- At the end of line 33 …oral administration of DNA-alkylating drug temozolomide, show limited efficacy in GBM treatment. Authors should indicate a reference!  

3- At the end of phrase   ….The microenvironment of GBM cells also plays an important role in the development, recurrence and resistance of brain cancer. In the lines 41 and 42, the authors should indicate a reference.  

4- In the lines 54 and 55 the authors wrote …Several subtypes (mainly containing α1, α7, α9 and α10 subunits) of nicotinic acetylcholine receptors (nAChRs) effectively permeate calcium ions [16]. Although it is not the target subject of this research, but without a doubt it is a very pertinent subject, I believe here that the authors could write something about extracellular microvesicles, e.g. Exosomes, ectosomes and oncosomes, which among others are known to function as signalosomes and are extremely dependent on the increase in intracellular calcium and surely are involved with development of glioblastomes.    

5- At the end of phrase   …. Several studies suggest that GBM proliferation, migration and invasion change in the presence of nAChR ligands. In the lines 64 and 65, the authors should indicate a reference.  

6- As well as in the lines 65 and 66 … These effects are predominantly mediat- 65 ed by α7 and α9 nAChRs. Please include a reference!  

7- As well as in the lines 67 and 68 … Complementary to that, 10-7 M nicotine increases the proliferation of the U87MG and GBM lines. Please include a reference!  

8- Also for sentence wrote between lines 73 and 76, please include a reference!  

9- Regarding the legend of figure 1, between lines 96 and 101, it is very enlightening, but I believe that the authors could mention the origins of the toxins used  ... peptide neurotoxins azemiopsin,[A10L]PnIA and RgIA are shown by their respective sequences, small protein α-bungarotoxin is shown by ribbon representation of spatial structure. From which venoms they were identified?  

10- In the line 104, 109 and 110… The Ivy Glioblastoma Atlas Project (Ivy GAP) database was used for the analysis.  Please include a reference or link to internet media!  

11- In the legend of figure 2 authors could change the definitions of colors. For instance, change color (electric Green), line 112 to green, (picton blue), line 112, change to blue.    

12- Regarding figure 2, I missed a more refined discussion about the reproducibility of the data. The authors indicate that the data was collected from an “Ivy Glioblastoma Atlas Project (Ivy GAP) database”. But I ask how representative these data are in relation to different patients? Could there be individual, geographic and racial variations in tumors? If so, could this lead to false interpretations of the data?  

13- About figure 3 legend, lines 142 to 145. In my opinion authors should change neurospheres to microspheres, because this phenomenon is not restricted to neuronal or brain cells! .  

14- Also figures of cells under FBS supplemented culture could be improved as possible, as they show a low number of cells for 011 cells,  067 cells, and U87MG, or in the case of 019 cells the focus of microphotography could be improved.  

15- Of course, it was not the goal of the study, but the profile of organization and distribution of cells in culture in the absence of fetal bovine serum (such as microspheres) indicates an increase in intracellular adhesions, in addition to metabolic cooperation, probably through gap junctions, and this could be studied evaluated, or commented.  

16- Also the behavior of the 022 cell sub lineage, which even in the presence of fetal bovine serum was shown to be in the form of microspheres deserves some discussion. Could this have any biological and molecular meaning? Probably yes!  

17- Still about figure 3, authors could think to show high magnification of figures, since as depicted some details could be lost. I really would like to see if under culture conditions used, cells produce extracellular vesicles or microvesicles as exosomes or ectosomes or oncosomes, which are signalosomes very important for cell biology and tumor behavior.   

18- In figure 4, it appears that the 067 cells were those that indicated lower levels of nAChR subunits expressed genes as pointed by figure 4a. Wouldn't these data be because these cells were the ones that apparently (from the data shown in figure 3) were the cells that grew least under the conditions used?  

19- However, the data shown in Figures 4b and 4c indicate that the transcripts for α5, α7 and α9 nAChR mRNAs were reasonably well expressed under the conditions tested in both the presence and absence of FCS, except for the α7 receptor in 019 cells. In this sense, the map shown in figure 4a is confusing. I would like some comments from the authors.  

20- Beteween lines 174 to 176 …. α-Neurotoxin α-Bgt from a multi-banded krait venom is a competitive antagonist of α7, α9 and muscle nA ChRs and binds to these receptors with high affinity. In my opinion authors should indicate the scientific name of multi-banded krait. Bungarus multicinctus or Bungarus spp?  

21- Regarding figure 5, initially it would have been good if the authors had double marked cells, using DAPI to visualize the nuclei. This would facilitate visualization of the cells as a whole and is a common procedure in manuscripts submitted with cell biology methodology.  

22- I understand the use of α-Neurotoxin α-Bgt to indicate the presence and viability of nAChR on the surface of the cells studied. But aren't there antibodies that recognize these receptors, which would be more enlightening and specific to confirming the data?  

23- In figure 5, some special reason to authors to have chosen 067 cells to perform immunofluorescence assays? Some details could be pointed by authors!  

24- Why other GMB cells were not used to study the phenotypic presence of nAChR in the models?  

25- To confirm GMB translation in the mesenchymal cell line (lines 185 and 186) authors could use specific antibodies to mesenchymal cells.  

26- In the lines 191 and 192 … Since mRNA expression of α1, α7 and α9 subunits forming calcium-conducting nAChRs was detected. A citation of a reference would make this sentence more complete.     

27- In the line 194 … Using calcium imaging technique,… Better to indicate which technique was or were used. Conventional fluorescence, confocal fluorescence, spectrofluorimetry or other?  

28- About figure 6, some special reason to authors to have chosen U87MG  cells to perform this assays? Some details could be pointed by authors!  

29- The lines of graphical pointed in figures 6 b, c, d, and e what means? Individual assays for the same experiment? Authors should include this information in the legend of figure!  

30- Also for figure 6a what was the time for registration of picture pointed? The peack of fluorescence 40 seconds?  

31- The author did this same experiments in the absence of FCS?  

32- For graphics pointed in figure 6b, 6c and 6d there were significant variations among lines depicted. Why authors interpret these differences? They are individual experiments of the same analysis?    

33- About figure 7, the authors wrote between lines 216 to 217 …. Similar experiments with inhibition of ACh-evoked Ca2+ entry were carried out  with patient-derived GBM cell lines. However the figure 7, shows data pointed in different manner of figure 6. Why reason authors changed the presentation of results? Any special reason?  

34- Given the data shown in Figure 7, wouldn't Figures 6b, c, d and e be repetitive and therefore could be removed from the revised version, since the experiments shown in Figure 7 include the data shown?  

35- In my opinion authors could divide figure 7 in at least two figures. The item 7D could be shown as figure 8 for instance.  

36- Again as previously pointed in figure 6, data shown in figure 7 showed variations for the same analysis! Why authors interpret these results?  

37- If authors do not divide figure 7 in two figures, then additional details about 7D must be included in the legend of figure!  

38- About figure 7D, it seems that it was another experiment. What means lines in red, green and blue?  

39- In figure 8, authors pointed results of experiments of influence of different culture conditions on the growth of cells of GMB. Why not also to study influences on migration of cells, as GMB tumors are prominent in metastasis and invasion?  

40- In the lines 257 and 258 authors wrote … GBM cell proliferation was studied using resazurin based colorimetric method (AlamarBlue). Some special reason to choose this kind of method among several other possibilities?  

41- Between lines 259 and 261 authors wrote  … In the AlamarBlue proliferation assay, non-selective nAChR agonist epibatidine was used in order to eliminate the influence on mAChR. Any special reason to do this procedure. Some detail could be included in the text!  

42-   Between lines 269 and 271 authors wrote    … Also the proliferation of U87MG grown on serum-free medium increased in response to azemiopsin in combination with epibatidine at 10 and 30 nM.  Some reason to use two drugs in combination?  Details could be included to facilitate readers interpretation!  

43- The experiments shown in Figure 8 (cell proliferation) in the presence or absence of FCS and nAChR ligands were tested. But after analyzing the results, the vast majority of treatments (especially serum free) did not influence the growth of different cells, or shown * p<0.05, indicating that these results need to be checked and repeated. Would I like the authors' opinions on this?  

44- With rare exceptions, the differences were small. Have the authors never considered using other techniques for proliferation assays, which perhaps show more significant differences?  

45- Also, have the authors never considered testing tumor growth in vivo?  

46- The authors wrote in lines 296 and 297  … Since addition of antagonists together with an agonist induced proliferation of GBM cells … It does not seem contradictory that antagonists together with agonists influenced positively tumor growth, or any other phenotypic aspect, since there must be competition for the same binding sites, but activation of different intracellular signaling and therefore, generations of different signals, since a drug is an agonist and another antagonist? Or activities are non canonical?  

47- When we see the results shown in figure 9, we realize that the increases in proliferation in different cells of GMB origin caused by exposure to nAChR antagonists are small or absent. And they are even more dependent on the presence of FCS. These data do not leave me convinced about the effective action of nAChR antagonists. Would I like to know the authors' opinion?  

48- Between lines 318 to 320 authros wrote …. In the experiments described above, the response to antagonists was higher in cells grown in FBS supplemented medium. This phenomenon may be explained by the effect of choline present in the serum. Unless the authors have carried out specific tests, this conclusion is very speculative, since serum contains thousands of substances that can act on different cells, and among these substances, dozens of them certainly have proliferative effects!  

49- Commentary wrote between lines 321 to 330 should be more appropriated to discussion chapter!  

50- To be sure of the proliferative activities postulated by the authors for nAChR modulators, I suggest that the authors look for a more sensitive proliferation detection method, where differences are noted with greater sensitivity, and without the presence of FCS, since fetal serum has hundreds of factors that can influence the test.  

51- Cells cultured in the absence of fetal serum would enter G0 in the cell cycle and would be synchronized and more sensitive to direct activities of agonists or antagonists influencing cell division.  

52- Regarding the experiments shown in figure 10, the biggest criticism we can make is that there was no concentration-dependent choline activity. Increases in choline concentrations did not give rise to concentration-dependent responses. This argues against some biological meaning of the data. I would like to know about opinion of authors related this fact?  

53- ... As already mentioned, line 067 is the most aggressive among the lines tested and its proliferation increased in response to the action of muscle, α7 and α9 nAChRs antagonists in a wide range of choline concentrations. If really there is competition between choline and antagonists, or if choline alone stimulates cell proliferation, the data should be concentration dependent. Either the experiments need to be improved, or the authors are in a range of non-responsive choline concentration. They should either increase concentration of choline, or decrease it.  

54- Discussion between lines 384 to 387 should be accompanied with a reference!  

55- In the line 389 … found in cancer cell lines repeatedly passaged in vitro often bear. Please include …as for instance repeatedly passes in vitro often bear …  

56- The discussion about problems observed in cell cultures, shown between lines 397 to 411, justifies the carrying out of experiments on tumor growth and metastasis in vivo.  

57- In the lines 417 to 419 the authors indicate figure 4    … Under different cell culture conditions (FBS supplemented vs. NS-21/EGF/bFGF-containing) patient-derived GBM cultures from different human sources have shown distinct profile of nAChR genes expression (Figure 4). But in figure 4 authors discuss nAChR genes expression in the presence or absence of FCS. They did not discuss in the text or legend about use of Growth Factors in the experiment!  

58- Between lines 431 to 434 the authors wrote …. The presence of fuctionally active nAChR in mesenchymal line 067 was confirmed by specific binding of fluorescent ligand Alexa Fluor 555 α-bungarotoxin (Figure 5), showing that nAChR genes expression led to the respective protein production by GBM cells. Although the authors showed fluorescent labeling of the cells studied, the conclusion is that the cells express toxin-binding proteins, it is not possible to conclude that they are active, since the experiment does not show inhibition/activation of activities conferred by the protein in question. So, I suggest removing from the text  … functionally active.  

59- Include a reference for text wrote between lines 435 to 437 … Functions of nAChRs in GBM cell cultures were studied using selective antagonists  α-conotoxins [A10L]PnIA and RgIA which bind preferably to α7 and α9 nAChRs, respectively.  

60- Include a reference for text wrote between lines 437 to 438    … Neurotoxin azemiopsin from Azemiops feae viper venom is known selectively block muscle α1β1δε nAChR.  

61- In the line 470, authors wrote …can reach concentration of several μM.  Better write a number for concentration of choline, although approximate. Several is undefined!  

62- In the lines 474 and 475   ….Choline is a metabolic precursor of the major endogenous cholinergic neurotransmitter acetylcholine and plays an important role in the nAChR function. Please include a reference for this affirmation!  

63-  In the lines 489 to 491 the authors wrote …No effect of nAChR ligands (antagonists or agonists) was detected on proliferation of proneural cell line 019 (Figure 8c,d) despite this line clearly showing the presence of functional nAChRs in calcium imaging and patch-clamp experiments. If the authors are indicating that nAChR receptors are important in activating cell proliferation, how can they explain this discrepancy? Could they be alternative intracellular signaling pathways and differences in gene regulation? Some comments about this will increase the attractively of text!  

64- From the analyzes carried out by qRTPCR for nAChR mRNA transcripts, it was clear that there were variations in expression in cultures in the presence and absence of FCS. I would like some discussion from the authors to interpret these data. How can the absence of FCS change the expression profile of these genes?  

65- It would be interesting for the authors to discuss possible intracellular signaling activated by nAChR ligands, and their relationship with activation of cell proliferation. Certainly the literature has already shown intracellular pathways activated after nAChR activation.

Author Response

General comments: Glioblastomas represent a type of cancer classified as Big Killers, due to their high capacity to kill affected patients. Any novelty that brings serious knowledge in this area, and that can serve to mitigate and alleviate the suffering of affected patients deserves respect and serious judgment. In this manuscript the authors are studying muscarinic acetylcholine receptors and the role of neurotoxins in these receptors presented in Glioblastoma multiforme lineage (U87MG GBM cell line) and primary cultivated cells obtained from patients with these tumors. Throughout the text, the authors report their findings, which show neutroxins that present activity on the subtypes of nicotinic acetylcholine receptors α1, α7 and α9, when exposed to Glioblastoma U87MG GBM cell lines, as well as cells obtained from patients with glioblastoma , and treated with nicotinic agonists or antagonists, showed proliferative activity on these cells under specific culture cell conditions. The data strengthen the use of primary cultures as models to study this type of tumor, as well as opening the way for learning about the cell biology of glioblastomas. After carefully reading the text, it is my opinion that this is a subject that falls within the scope of TOXINS, a subject of interest in public health, and finally a research that describes the participation of cellular acetylcholine receptors influencing tumor cell growth. Attached are some suggestions that could make a modified version of the text more complete and attractive.

We thank the Reviewer for the comprehensive review of the manuscript. We believe that our work could be interesting to the colleagues working with nAChR and tumor cells. We tried our best to address all suggestions where it was possible. Several supplementary files were added, some figures were updated and text was ammended.

Specific Comments:

1- In the line 25 the authors wrote…. …shown that nAChRs subtype-selective antagonists increase proliferation … In my opinion it is better to write the name of antagonists, as there are several types that could act in different ways.

We have added names of selective antagonists (line 25).

2- At the end of line 33 …oral administration of DNA-alkylating drug temozolomide, show limited efficacy in GBM treatment. Authors should indicate a reference!

We have added a reference (line 33).

3- At the end of phrase ….The microenvironment of GBM cells also plays an important role in the development, recurrence and resistance of brain cancer. In the lines 41 and 42, the authors should indicate a reference.

We have added a reference (line 42). 

4- In the lines 54 and 55 the authors wrote …Several subtypes (mainly containing α1, α7, α9 and α10 subunits) of nicotinic acetylcholine receptors (nAChRs) effectively permeate calcium ions [16]. Although it is not the target subject of this research, but without a doubt it is a very pertinent subject, I believe here that the authors could write something about extracellular microvesicles, e.g. Exosomes, ectosomes and oncosomes, which among others are known to function as signalosomes and are extremely dependent on the increase in intracellular calcium and surely are involved with development of glioblastomes.

Extracellular microvesicles are involved in cross-talk between glioblastoma cells and various cellular components of tumor microenvironment and, furthermore, are associated with several tumor-promoting mechanisms (doi: 10.1038/s41388-020-1308-2). However, despite the prominent role of extracellular microvesicles in glioblastoma progression, this topic is beyond the scope of this paper.

5- At the end of phrase …. Several studies suggest that GBM proliferation, migration and invasion change in the presence of nAChR ligands. In the lines 64 and 65, the authors should indicate a reference.

We have added a reference (line 65).

6- As well as in the lines 65 and 66 … These effects are predominantly mediat- 65 ed by α7 and α9 nAChRs. Please include a reference!

We have added a reference (line 66).

7- As well as in the lines 67 and 68 … Complementary to that, 10-7 M nicotine increases the proliferation of the U87MG and GBM lines. Please include a reference!

We have added a reference (line 68).

8- Also for sentence wrote between lines 73 and 76, please include a reference!

We have added a reference (line 75).

9- Regarding the legend of figure 1, between lines 96 and 101, it is very enlightening, but I believe that the authors could mention the origins of the toxins used  ... peptide neurotoxins azemiopsin,[A10L]PnIA and RgIA are shown by their respective sequences, small protein α-bungarotoxin is shown by ribbon representation of spatial structure. From which venoms they were identified?

Figure legend were modified according to the Reviewer suggestions.

10- In the line 104, 109 and 110… The Ivy Glioblastoma Atlas Project (Ivy GAP) database was used for the analysis.  Please include a reference or link to internet media!

We have added a reference (line 104).

11- In the legend of figure 2 authors could change the definitions of colors. For instance, change color (electric Green), line 112 to green, (picton blue), line 112, change to blue.

We have changed the definitions of colors: blue-green to dark blue (line 114), deep magenta to magenta (line 115), electric green to green (line 115), picton blue to light blue (line 115), electric orange to red (line 117).

12- Regarding figure 2, I missed a more refined discussion about the reproducibility of the data. The authors indicate that the data was collected from an “Ivy Glioblastoma Atlas Project (Ivy GAP) database”. But I ask how representative these data are in relation to different patients? Could there be individual, geographic and racial variations in tumors? If so, could this lead to false interpretations of the data?

Briefly, the database contain anatomic-aware data on gene expression in 42 glioblastoma tumors from 41-patient cohort. This is not particularly large sample but it contains simultaneous detection of gene expression with respect to the tumor morphology, which helps to gain insight on possible functions of nAChR genes in GBM.

13- About figure 3 legend, lines 142 to 145. In my opinion authors should change neurospheres to microspheres, because this phenomenon is not restricted to neuronal or brain cells! .

The term “neurosphere” implies spherical clusters of cells enriched with neural stem cells (NSCs).It has been suggested that gliomas (including glioblastomas) might originate from neural stem cells as well as glial progenitor cells (DOI: 10.1056/NEJMra043666). Neurosphere formation in culture is one of defining characteristics of this type of brain tumor and, therefore, the terms “neurosphere” and “neurosphere culture” are frequently applied while describing cell cultures of gliomas and glioblastomas (doi: 10.1002/stem.15; doi: 10.1007/s11060-019-03107-0).

However, we acknowledge the Reviewer point of view and have changed neuro- to microspheres throughout the manuscript.

14- Also figures of cells under FBS supplemented culture could be improved as possible, as they show a low number of cells for 011 cells, 067 cells, and U87MG, or in the case of 019 cells the focus of microphotography could be improved.

We tried to address this issue and changed pictures of FBS cultured cells on Fig. 3.

15- Of course, it was not the goal of the study, but the profile of organization and distribution of cells in culture in the absence of fetal bovine serum (such as microspheres) indicates an increase in intracellular adhesions, in addition to metabolic cooperation, probably through gap junctions, and this could be studied evaluated, or commented.

Thank you for the wonderful idea. We will try to pay attention to it in the new study. As you correctly pointed out, this was not the purpose of the study in this paper. Numerous publications have also been devoted to growth factors and they are considered therapeutic targets, since they affect the epithelial - mesenchymal transition. https://www.ncbi.nlm.nih.gov/pmc/articles/PMC4774466 / In our work, we investigated the lines of the proneuronal and mesenchymal phenotype.

16- Also the behavior of the 022 cell sub lineage, which even in the presence of fetal bovine serum was shown to be in the form of microspheres deserves some discussion. Could this have any biological and molecular meaning? Probably yes!

Yes, indeed, line 022 does not form an adhesive monolayer in a medium containing serum. Moreover, this cell line most often does not survive on a medium with serum. We will definitely study this effect and describe it in subsequent works.

17- Still about figure 3, authors could think to show high magnification of figures, since as depicted some details could be lost. I really would like to see if under culture conditions used, cells produce extracellular vesicles or microvesicles as exosomes or ectosomes or oncosomes, which are signalosomes very important for cell biology and tumor behavior.

We appreciate the reviewer’s suggestion and agree that higher magnification and higher resolution of microphotographs would reveal more details. However, the microscopy techniques available in our department do not allow us to obtain more detailed images.

18- In figure 4, it appears that the 067 cells were those that indicated lower levels of nAChR subunits expressed genes as pointed by figure 4a. Wouldn't these data be because these cells were the ones that apparently (from the data shown in figure 3) were the cells that grew least under the conditions used?

Line 067 grows reasonably well in FBS-supplemented and serum-free medium. We corrected figure 3 to show better the cell count. In fact, figure 4a shows relative change in nAChR gene expression in serum free versus FBS-supplemented medium. Thus, line 067 did not changed expression of CHRNA5, CHRNA7 and CHRNA9, but all these genes were on the level comparable to other lines. We corrected figure 4 to improve data representation.

19- However, the data shown in Figures 4b and 4c indicate that the transcripts for α5, α7 and α9 nAChR mRNAs were reasonably well expressed under the conditions tested in both the presence and absence of FCS, except for the α7 receptor in 019 cells. In this sense, the map shown in figure 4a is confusing. I would like some comments from the authors.

We thank the Reviewer for rising this concern. Color-coding of the heatmaps was confusing, and we missed this fact during the manuscript preparation. We have changed figure 4 to reflect better the actual gene expression pattern. Figure legend was also amended.

20- Beteween lines 174 to 176 …. α-Neurotoxin α-Bgt from a multi-banded krait venom is a competitive antagonist of α7, α9 and muscle nA ChRs and binds to these receptors with high affinity. In my opinion authors should indicate the scientific name of multi-banded krait. Bungarus multicinctus or Bungarus spp?  

We agree and have added the scientific name Bungarus multicinctus to the text (line 177).

21- Regarding figure 5, initially it would have been good if the authors had double marked cells, using DAPI to visualize the nuclei. This would facilitate visualization of the cells as a whole and is a common procedure in manuscripts submitted with cell biology methodology.

We agree with the Reviewer and the necessary pictures have been added to the Fig. 5 panels “C” and “D” to facilitate cells visualization.

22- I understand the use of α-Neurotoxin α-Bgt to indicate the presence and viability of nAChR on the surface of the cells studied. But aren't there antibodies that recognize these receptors, which would be more enlightening and specific to confirming the data?

This is very interesting topic that deserves separate discussion. If possible, we would be glad to avoid this discussion in the current paper which is already inflated with various facts. In our opinion it is largely accepted within nAChR community that commercially available nAChR-targeted antibodies are not selective. Some scientific groups developed good antibodies. For example, authors personal communications with group of Prof. Kummer from Justus-Liebig University in Gieβen (Germany) confirm that such antibodies could be used with high level of credibility. However, most of nAChR antibodies are notoriously bad and all such results should be validated with fluorescent-labeled neurotoxins anyway.

As an alternative to antibodies, authors are now developing proteomics approach that allows to measure simultaneous Cys-loop (nAChRs and other) receptors detection. These results, however, are preliminary and we would prefer not to disclose them in current manuscript, if the Reviewer would not insist. We believe that combination of qPCR, calcium imaging, patch-clamp, fluorescent microscopy and alamarBlue assay demonstrates presence of nAChR sub-types in our primary GBM lines resonably well.

23- In figure 5, some special reason to authors to have chosen 067 cells to perform immunofluorescence assays? Some details could be pointed by authors!

The 067 cell culture is the most aggressive of all patient-derived cultures studied in this work, thus it is of a great clinical interest. These cells are also showing better adhesion to the coverslip which is used in our confocal microscopy set-up, thus they were our best object in this experiment.

To address this and the next question we add pictures of 019 GBM cell line (proneural type).

24- Why other GMB cells were not used to study the phenotypic presence of nAChR in the models?

As said before, 067 line was more interesting because among our GBM lines it is the most aggressive. The functional activity of nAChRs all cell cultures was studied by fluorescent calcium imaging technique, demonstrating to what extent functional nAChR are expressed on the cell surface. But fluorescent toxin staining gained resonably well results only on lines 067 and 019, we believe, due to technical reasons such as bad adhesion to the glass coverslips.

25- To confirm GMB translation in the mesenchymal cell line (lines 185 and 186) authors could use specific antibodies to mesenchymal cells.

We attach qPCR results showing proneuronal (CD133, Sox2, Olig2) and mesenchymal (CD44, ALDH1A3, Met) subtype markers expression as supplementary figure S4.

26- In the lines 191 and 192 … Since mRNA expression of α1, α7 and α9 subunits forming calcium-conducting nAChRs was detected. A citation of a reference would make this sentence more complete.

We have added a reference to Figure 4 (line 197).

27- In the line 194 … Using calcium imaging technique,… Better to indicate which technique was or were used. Conventional fluorescence, confocal fluorescence, spectrofluorimetry or other?

We have added a phase “at the epifluorescent microscope” to address this reviewer’s suggestion (line 199).  

28- About figure 6, some special reason to authors to have chosen U87MG  cells to perform this assays? Some details could be pointed by authors!

Fluorescent calcium imaging assay was performed on all of the GBM cell lines included in this study. Figure 6 depicts the characteristic transient fluorescent signals from the cytoplasmic calcium indicator Fluo-4 AM before and after the exposure of nAChR ligands. We decided to plot only one cell line as an example of the recorded data, since adding all of the cell lines cultivated in both serum-free and serum-supplemented media to one figure makes it much more difficult for perception. The model line U87MG was chosen as an example.

29- The lines of graphical pointed in figures 6 b, c, d, and e what means? Individual assays for the same experiment? Authors should include this information in the legend of figure!

As noted in the caption to the Figure 6, each individual line shows the change in intracellular Ca2+ concentration in a single cell.

30- Also for figure 6a what was the time for registration of picture pointed? The peack of fluorescence 40 seconds?

The total recording time was 70 seconds. The upper microphotograph shows the time point 0 seconds, while the bottom depicts the moment when the signal intensity reaches its maximum.

31- The author did this same experiments in the absence of FCS?

Fluorescent calcium imaging assay was performed on all of the GBM cell lines included in this study in both serum-free and serum-supplemented media.

32- For graphics pointed in figure 6b, 6c and 6d there were significant variations among lines depicted. Why authors interpret these differences? They are individual experiments of the same analysis?

As noted in the caption to the Figure 6, each individual line shows the change in intracellular Ca2+ concentration in a single cell. The glioblastoma cell cultures studied in this research are heterogeneous, consequently, differences in the amplitude of recorded signals between cells within the same cell culture were observed. For further statistical analysis we used the average amplitude of transient signals. Non-parametric statistical tests were used to

33- About figure 7, the authors wrote between lines 216 to 217 …. Similar experiments with inhibition of ACh-evoked Ca2+ entry were carried out  with patient-derived GBM cell lines. However the figure 7, shows data pointed in different manner of figure 6. Why reason authors changed the presentation of results? Any special reason?

Figure 7 displays the amplitudes of transient fluorescent signals in response to ACh application. It was calculated as the difference between the maximum fluorescence intensity of the calcium indicator Fluo-4 AM and the fluorescence intensity at time point 0 seconds. Violin plots were chosen for data visualization because this type of graph allow us to visually compare the distribution of amplitudes after the exposure to different combinations of nAChR ligands. 

34- Given the data shown in Figure 7, wouldn't Figures 6b, c, d and e be repetitive and therefore could be removed from the revised version, since the experiments shown in Figure 7 include the data shown?

Figures 6b, c, d and e show the change in fluorescence intensity of the calcium indicator Fluo-4 AM after the application of the nAChR agonist acetylcholine (ACh), whereas Figure 7a, b, c, e, f depict the calculated amplitudes of transient fluorescent signals and statistical analysis of obtained data.  

35- In my opinion authors could divide figure 7 in at least two figures. The item 7D could be shown as figure 8 for instance.

We appreciate the reviewer’s suggestion, however, we believe that adding a new figure, as the reviewer advised, would be unnecessary since we added the description of Figure 7D. 

36- Again as previously pointed in figure 6, data shown in figure 7 showed variations for the same analysis! Why authors interpret these results?

Figures 6b, c, d and e show the change in fluorescence intensity of the calcium indicator Fluo-4 AM after the application of the nAChR agonist acetylcholine (ACh), whereas Figure 7a, b, c, e, f depict the calculated amplitudes of transient fluorescent signals and statistical analysis of obtained data. 

37- If authors do not divide figure 7 in two figures, then additional details about 7D must be included in the legend of figure!

We would like to thank the reviewer for bringing this oversight to our attention. We have extended the description of Figure 7.

38- About figure 7D, it seems that it was another experiment. What means lines in red, green and blue?

Figure 7D shows concentration-dependent ion currents in whole-cell patch-clamp experiment for 019 cell culture grown in FBS-supplemented medium in response to mixture of α7 nAChR-selective agonist PNU 282987 and positive modulator PNU 120596. Lines of different colors show different concentrations of the α7 nAChR-selective agonist: red (1 μM), green (10 μM) and blue (100 μM).

39- In figure 8, authors pointed results of experiments of influence of different culture conditions on the growth of cells of GMB. Why not also to study influences on migration of cells, as GMB tumors are prominent in metastasis and invasion?

Indeed, at the moment we are studying the effect of nAChR ligands on the cell migration of primary GBM cultures. This data will be included in the next publication.

40- In the lines 257 and 258 authors wrote … GBM cell proliferation was studied using resazurin based colorimetric method (AlamarBlue). Some special reason to choose this kind of method among several other possibilities?

We chose AlamarBlue because the cell cycle time of primary glioblastoma cultures has been increased. To fully assess the effect of the ligands, the measurement was performed on the 5th day, which, for example, is impossible for the MTT test. Moreover, the appeal of AlamarBlue is that it is a vital test (not leading to cell death).

41- Between lines 259 and 261 authors wrote … In the AlamarBlue proliferation assay, non-selective nAChR agonist epibatidine was used in order to eliminate the influence on mAChR. Any special reason to do this procedure. Some detail could be included in the text!

This paper focused on the activity of nAChRs, so it was necessary to use agonists that activate only nAChRs and do not act on another type of acetylcholine receptors, mAChRs. Epibatidine activates only nAChRs and, therefore, it was chosen as an agonist in AlamarBlue proliferation assay.

42- Between lines 269 and 271 authors wrote … Also the proliferation of U87MG grown on serum-free medium increased in response to azemiopsin in combination with epibatidine at 10 and 30 nM. Some reason to use two drugs in combination? Details could be included to facilitate readers interpretation!

Epibatidine is a nAChR agonist, whereas azemiopsin is a selective muscle-type receptor inhibitor. In proliferation tests, the effects of both agonist-mediated activation of receptors and their inhibition by selective antagonists were investigated. The addition of an antagonist in combination with an agonist is necessary to investigate whether the antagonist counteracts possible effects of the agonist. Manuscript was amended to reflect this idea.

43- The experiments shown in Figure 8 (cell proliferation) in the presence or absence of FCS and nAChR ligands were tested. But after analyzing the results, the vast majority of treatments (especially serum free) did not influence the growth of different cells, or shown * p<0.05, indicating that these results need to be checked and repeated. Would I like the authors' opinions on this?

Well, we were also disappointed when saw the results for the first time. We anticipated to detect some excitotoxicity of epibatidine due to uncontrolled activation of the ion channels and toxins rescuing cells via ion channels inhibition. That’s not what we observed. However, these results demonstrated i) that effects of nAChR ligands on cells viability are very limited (which was not obvious, because calcium imaging shows strong effects of nAChR agonists on intracellular calcium); ii) effects depend on the cell line, thus showing the heterogenity of GBM with respect to nAChR action; iii) effects could depend on culturing medium selection; iv) epibatidine did not evoke excitotoxicity on GBM, but simultanious application of antagonists could stimulate cell growth (so the experiment shown on figure 9 was set up).

Although disappointing, this experiment honestly shows the results of chronic treatment of the cells with epibatidine.

44- With rare exceptions, the differences were small. Have the authors never considered using other techniques for proliferation assays, which perhaps show more significant differences?

We tested Ki-67 immunostaining, but the effects are still small or nonexistent. Really, the small size of the effects is the main finding here. We understand that it does not sound exciting, but we believe that this information could be beneficial to our colleagues who plan their experiments with nAChR ligands.

45- Also, have the authors never considered testing tumor growth in vivo?

To date we have optimized murine models of subcutaneous and brain xenografts for some of our lines (plus U87MG that serves a standard in such in vivo tests). Proteomes of these xenografts are analyzed now, but the results are far from conclusive and further work will be done before we could publish these results.

46- The authors wrote in lines 296 and 297  … Since addition of antagonists together with an agonist induced proliferation of GBM cells … It does not seem contradictory that antagonists together with agonists influenced positively tumor growth, or any other phenotypic aspect, since there must be competition for the same binding sites, but activation of different intracellular signaling and therefore, generations of different signals, since a drug is an agonist and another antagonist? Or activities are non canonical?

The addition of antagonists together with agonists is necessary to investigate whether the antagonist counteracts possible effects of the agonist, since agonists and antagonists used in this study compete for the same orthosteric binding site. However, the results obtained not readily fit into such straightforward model. The main idea here is that cells are adapting to chronic administration of nAChR agonists by down-regulating the receptors, whereas addition of the toxins counteracts this rundown by limiting the receptor activation. While cells are not visibly depressed with chronic agonist administration, nAChR inhibition could play in favor of cells by removing this stress-factor, thus slightly stimulating cells viability.

47- When we see the results shown in figure 9, we realize that the increases in proliferation in different cells of GMB origin caused by exposure to nAChR antagonists are small or absent. And they are even more dependent on the presence of FCS. These data do not leave me convinced about the effective action of nAChR antagonists. Would I like to know the authors' opinion?

In this experiment we hypothesised the chronic nAChR activation in FBS-supplemented medium. Since previous experiments showed slight increase in proliferation, we simplified the experimental design and excluded the epibatidine factor. Indeed, without additional nAChR activation by epibatidine, increase in proliferation is observed only in FBS-supplemented medium. The effects are still small, reaching 50% only in model U87MG cell line, but are statistically significant. We do not tend to overestimate these effects, but the results are reported fairly and could be used to infer the properties of toxins in regards to GBM viablility. What we definitely do not see is the inhibition of the cell growth reported previously which would be interesting to know to our colleagues.

48- Between lines 318 to 320 authros wrote …. In the experiments described above, the response to antagonists was higher in cells grown in FBS supplemented medium. This phenomenon may be explained by the effect of choline present in the serum. Unless the authors have carried out specific tests, this conclusion is very speculative, since serum contains thousands of substances that can act on different cells, and among these substances, dozens of them certainly have proliferative effects!

FBS contains thousands of compounds, indeed. In some sense it is a “black box”. But not all of these compounds bind and activate nAChRs. Choline is certainly present in FBS and is certainly activate alpha1, alpha7 and alpha9. Thus it does not seem completely wild guess to hypothesize such an action of FBS. But we understand that even in such case a plethora of other compounds from FBS can modulate choline action. We remove the questionable phrase to avoid confusion.

49- Commentary wrote between lines 321 to 330 should be more appropriated to discussion chapter!

We thank the reviewer for this suggestions and move this commentary to the discussion section.

50- To be sure of the proliferative activities postulated by the authors for nAChR modulators, I suggest that the authors look for a more sensitive proliferation detection method, where differences are noted with greater sensitivity, and without the presence of FCS, since fetal serum has hundreds of factors that can influence the test.

You are right that fetal serum and hundreds of factors can affect the test result. But the test was conducted under standard conditions and we only note the trend in these specific conditions using this method. But we agree with your approach and will devote a separate publication to the study of subtle influences.

51- Cells cultured in the absence of fetal serum would enter G0 in the cell cycle and would be synchronized and more sensitive to direct activities of agonists or antagonists influencing cell division.

We conducted experiments on synchronization of primary cell cultures. It turns out to be too difficult for glioblastoma and the percentage of synchronized cells is small, so we did not continue these experiments on primary cultures, because there are a lot of artifacts in their results. We cultivate without serum, but with growth factors and cells are not entering the G0 cell cycle. We tried cultivating without some factors (canceling B and E supplements), but there is also no such effect.

52- Regarding the experiments shown in figure 10, the biggest criticism we can make is that there was no concentration-dependent choline activity. Increases in choline concentrations did not give rise to concentration-dependent responses. This argues against some biological meaning of the data. I would like to know about opinion of authors related this fact?

Very much like in the case of epibatidine experiment (figure 8) nAChR agonist, in this case choline, did not visibly influence cell viablility. However, addition of the antagonists to lines 011, 019 and 067 gave significant increase in cell proliferation which was not observed in serum-free conditions whithout choline.

The concentration dependence is not monotonous only in case of the line 019 and was not detected in U87 and 022 line at all. However, in line 011we see the increase in effects of RgIA and PnIA with the choline concentration increase. In case of 067 line quiet the opposite happens with the effects of RgIA and PnIA. Different effects of the higher choline concentrations are not completely surprising. Choline-sensitive nAChRs (muscle, alpha7 and alpha9) are known to desensitize at higher concentrations of agonists (alpha7 being the fastest desensitizing among them). Thus, giving slightly different nAChR subsets expressed in these lines, complex combination of the receptor cross-talk, inhibition and desensitization can give rise to non-linear responses.

That seems to be a serious topic and we are contacting mathematics department to develop differential equations-based model to study such phenomenon further.

53- ... As already mentioned, line 067 is the most aggressive among the lines tested and its proliferation increased in response to the action of muscle, α7 and α9 nAChRs antagonists in a wide range of choline concentrations. If really there is competition between choline and antagonists, or if choline alone stimulates cell proliferation, the data should be concentration dependent. Either the experiments need to be improved, or the authors are in a range of non-responsive choline concentration. They should either increase concentration of choline, or decrease it.

The range between 100 nM and 100 microM definitely covers what we can encounter in 10% FBS supplemented medium. Moreover, 100 microM is sufficient to activate muscle, alpha7 and alpha9 nAChR according to published data (100 nM is by far less than needed to observe receptor activation). We understand that the effects on cell viability are far from exciting. But this is honest data and concentration range was selected thoughtfully.

Considering the competition between choline and toxins, we should keep in mind the receptors cross-talk and desensitization under chronic choline exposure. It is virtually not possible to explore all space of possibilities within one experiment.

54- Discussion between lines 384 to 387 should be accompanied with a reference!

Reference 32 added.

55- In the line 389 … found in cancer cell lines repeatedly passaged in vitro often bear. Please include …as for instance repeatedly passes in vitro often bear …

We thank the Reviewer for this note. The respective text was ammended.

56- The discussion about problems observed in cell cultures, shown between lines 397 to 411, justifies the carrying out of experiments on tumor growth and metastasis in vivo.

We agree with this note. As said in answer to p.45 we are now optimizing the xenograft growth conditions. Some material has been gathered and waiting for proteomics research.

57- In the lines 417 to 419 the authors indicate figure 4 … Under different cell culture conditions (FBS supplemented vs. NS-21/EGF/bFGF-containing) patient-derived GBM cultures from different human sources have shown distinct profile of nAChR genes expression (Figure 4). But in figure 4 authors discuss nAChR genes expression in the presence or absence of FCS. They did not discuss in the text or legend about use of Growth Factors in the experiment!

We thank the Reviewer for finding this inconsistency in the text. By serum-free-medium we mean Medium II (see matherials and methods). We explained it in section 2.2, but we adding another note to figure 4 legend.

58- Between lines 431 to 434 the authors wrote …. The presence of fuctionally active nAChR in mesenchymal line 067 was confirmed by specific binding of fluorescent ligand Alexa Fluor 555 α-bungarotoxin (Figure 5), showing that nAChR genes expression led to the respective protein production by GBM cells. Although the authors showed fluorescent labeling of the cells studied, the conclusion is that the cells express toxin-binding proteins, it is not possible to conclude that they are active, since the experiment does not show inhibition/activation of activities conferred by the protein in question. So, I suggest removing from the text … functionally active.

We have accepted the reviewer's recommendation and have removed the phrase “functionally active” from this sentence (line 444).

59- Include a reference for text wrote between lines 435 to 437 … Functions of nAChRs in GBM cell cultures were studied using selective antagonists  α-conotoxins [A10L]PnIA and RgIA which bind preferably to α7 and α9 nAChRs, respectively.

We have added a reference (line 450).

60- Include a reference for text wrote between lines 437 to 438 … Neurotoxin azemiopsin from Azemiops feae viper venom is known selectively block muscle α1β1δε nAChR.

We have added a reference (line 451).

61- In the line 470, authors wrote …can reach concentration of several μM. Better write a number for concentration of choline, although approximate. Several is undefined!

We agree with reviewer’s observation. We have revised this paragraph (line 484).

62- In the lines 474 and 475….Choline is a metabolic precursor of the major endogenous cholinergic neurotransmitter acetylcholine and plays an important role in the nAChR function. Please include a reference for this affirmation!

We have added a reference (line 490).

63- In the lines 489 to 491 the authors wrote …No effect of nAChR ligands (antagonists or agonists) was detected on proliferation of proneural cell line 019 (Figure 8c,d) despite this line clearly showing the presence of functional nAChRs in calcium imaging and patch-clamp experiments. If the authors are indicating that nAChR receptors are important in activating cell proliferation, how can they explain this discrepancy? Could they be alternative intracellular signaling pathways and differences in gene regulation? Some comments about this will increase the attractively of text!

We agree with the Reviewer and added cooments regarding PKC and MAPK activation in response to nAChR ligands.

64- From the analyzes carried out by qRTPCR for nAChR mRNA transcripts, it was clear that there were variations in expression in cultures in the presence and absence of FCS. I would like some discussion from the authors to interpret these data. How can the absence of FCS change the expression profile of these genes?

It has already been shown that growing cells in a serum medium leads to their irreversible differentiation and loss of tumor stem cell populations. On the contrary, growing cells on a medium that does not contain serum, but contains growth factors, allows you to preserve the initial heterogeneity of the tumor and stem cells. In this regard, the gene expression profiles of cells grown under these two different conditions will differ and this has already been shown: https://www.cell.com/cancer-cell/fulltext/S1535-6108(06)00117-6

65- It would be interesting for the authors to discuss possible intracellular signaling activated by nAChR ligands, and their relationship with activation of cell proliferation. Certainly the literature has already shown intracellular pathways activated after nAChR activation.

It is widely accepted, that nAChR-mediated effects are dependent on protein kinase C activation and subsequent mitogen-activated protein kinases signal cascade [16]. Thus, exact reactions of the GBM to nAChR ligands should depend not only on nAChR subtipes surface expression, but also on pre-existance of such mechanisms in GBM cells.

Reviewer 3 Report

Comments and Suggestions for Authors

A very neatly done study comparing the effects of nAChR agonists and antagonists on different glioblastoma cell lines and looking at the effect of tumour progression. The experiments are very well designed and conclusions are very clear

Author Response

We thank the Reviewer for such positive response! The manuscript has been updated according to other reviewers. We hope that the manuscript has become better.

Round 2

Reviewer 1 Report

Comments and Suggestions for Authors

The authors have addressed most of my concerns and have edited the manuscript and figures appropriately.

Interestingly, to resolve the important question of the selectivity of the effect observed with the studied antagonists, the authors performed further experiments using Ca fluorescence and cytotoxicity assays reported in supplementary data. However, these results are not discussed in the manuscript, neither in the Results, nor in the Discussion. This must be done

Author Response

We thank the Reviewer for pointing out this inconsistency in the manuscript. We adding the following paragraphs to the text:

Lines 260-269 "To explore the selectivity of the effects observed in calcium imaging experiments the wider panel of nAChR ligands was deployed on 022 line (Figure S2). Interestingly, nicotine, PNU-282987, methyl ester of D-6-Iodohypaphorine (6ID) and D-6-nitrohypahporine amide (6NAM) evoked calcium responses. D-tubocurarune effectively inhibited all responses to solo agonists, but was not effective against PNU-282987 and 6NAM when co-applied with positive modulator PNU-120596 (pnu12). Three-finger toxin α-cobratoxin (α-CTX), being selective muscle, α7 and α9 nAChR inhibitor, diminished responses to nicotine and 6NAM when PNU-120596 was not co-applied. These results speak in favor of α7 nAChR presence in 022 line and are in good agreement with data presented on Fig. 7 e."

Lines 328-333 "Since epibatidine is an agonist of a broad spectrum of nAChR sub-types, selective α7 nAChR agonist PNU-282987 and α-CTX were also tested on 022 line to check the effect selectivity. In accordance to epibatidine/toxins results presented on Fig.8, no significant influence of PNU-282987 on 022 cell line proliferation has been observed at concentrations from 20 nM to 6 μM. Addition of 1 μM α-CTX to cells incubated with PNU-282987 increases cell proliferation. The effect is dose dependent and is observed only at low agonist concentrations (Fig. S3)."